# VDT: General-purpose Video Diffusion Transformers via Mask Modeling

**Haoyu Lu**[1] **Guoxing Yang**[1] **Nanyi Fei**[1] **Yuqi Huo**[4] **Zhiwu Lu**[1,*] **Ping Luo**[3] **Mingyu Ding**[2,*]

[1]Gaoling School of Artificial Intelligence, Renmin University of China, Beijing, China

[2]University of California, Berkeley, United States

[3]The University of Hong Kong, Pokfulam, Hong Kong

[4]Baichuan Inc.

{lhy1998, luzhiwu}@ruc.edu.cn    myding@berkeley.edu

## Abstract

This work introduces Video Diffusion Transformer (VDT), which pioneers the use of transformers in diffusion-based video generation. It features transformer blocks with modularized temporal and spatial attention modules to leverage the rich spatial-temporal representation inherited in transformers. Additionally, we propose a unified spatial-temporal mask modeling mechanism, seamlessly integrated with the model, to cater to diverse video generation scenarios.

VDT offers several appealing benefits. **1)** It excels at capturing temporal dependencies to produce temporally consistent video frames and even simulate the physics and dynamics of 3D objects over time. **2)** It facilitates flexible conditioning information, *e.g.*, simple concatenation in the token space, effectively unifying different token lengths and modalities. **3)** Pairing with our proposed spatial-temporal mask modeling mechanism, it becomes a general-purpose video diffuser for harnessing a range of tasks, including unconditional generation, video prediction, interpolation, animation, and completion, etc. Extensive experiments on these tasks spanning various scenarios, including autonomous driving, natural weather, human action, and physics-based simulation, demonstrate the effectiveness of VDT. Additionally, we present comprehensive studies on how VDT handles conditioning information with the mask modeling mechanism, which we believe will benefit future research and advance the field. Codes and models are available at VDT-2023.github.io.

## 1 Introduction

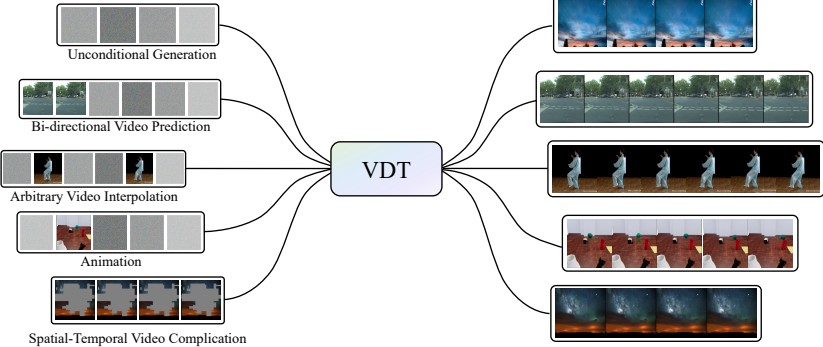

Figure 1: A diagram of our unified video diffusion transformer (VDT) via spatial-temporal mask modeling. VDT represents a versatile framework constructed upon pure transformer architectures.

Recent years have witnessed significant achievements in artificial intelligence generated content (AIGC), where diffusion models have emerged as a central technique being extensively studied in image (Nichol & Dhariwal, 2021; Dhariwal & Nichol, 2021) and audio domains (Kong et al.,

---
*Corresponding authors.

2020; Huang et al., 2023). For example, methods like DALL-E 2 (Ramesh et al., 2022) and Stable Diffusion (Rombach et al., 2022) can generate high-quality images given textual description. However, diffusion approaches in the video domain, while attracting a lot of attention, still lag behind. The challenges lie in effectively modeling temporal information to generate temporally consistent high-quality video frames, and unifying a variety of video generation tasks including unconditional generation, prediction, interpolation, animation, and completion, as shown in Figure 1.

Recent works (Voleti et al., 2022; Ho et al., 2022b;a; Yang et al., 2022; He et al., 2022; Wu et al., 2022a; Esser et al., 2023; Yu et al., 2023b; Wang et al., 2023b) have introduced video generation and prediction methods based on diffusion techniques, where U-Net (Ronneberger et al., 2015) is commonly adopted as the backbone architecture. Few studies have shed light on diffusion approaches in the video domain with alternative architectures. Considering the exceptional success of the transformer architecture across diverse deep learning domains and its inherent capability to handle temporal data, we raise a question: *Is it feasible to employ vision transformers as the backbone model in video diffusion?* Transformers have been explored in the domain of image generation, such as DiT (Peebles & Xie, 2022) and U-ViT (Bao et al., 2022), showcasing promising results. When applying transformers to video diffusion, several unique considerations arise due to the temporal nature of videos.

Transformers offer several advantages in the video domain. **1)** The domain of video generation encompasses a variety of tasks, such as unconditional generation, video prediction, interpolation, and text-to-image generation. Prior research (Voleti et al., 2022; He et al., 2022; Yu et al., 2023b; Blattmann et al., 2023) has typically focused on individual tasks, often incorporating specialized modules for downstream fine-tuning. Moreover, these tasks involve diverse conditioning information that can vary across frames and modalities. This necessitates a robust architecture capable of handling varying input lengths and modalities. The integration of transformers can facilitate the seamless unification of these diverse tasks. **2)** Transformers, unlike U-Net which is designed mainly for images, are inherently capable of capturing long-range or irregular temporal dependencies, thanks to their powerful tokenization and attention mechanisms. This enables them to better handle the temporal dimension, as evidenced by superior performance compared to convolutional networks in various video tasks, including classification (Wang et al., 2022b; 2023a), localization (Zhang et al., 2022; Wang et al., 2023a), and retrieval (Wang et al., 2022a; Lu et al., 2022). **3)** Only when a model has learned (or memorized) worldly knowledge (*e.g.*, spatiotemporal relationships and physical laws) can it generate videos corresponding to the real world. Model capacity is thus a crucial component for video diffusion. Transformers have proven to be highly scalable, making them more suitable than 3D-U-Net (Ho et al., 2022b; Blattmann et al., 2023; Wang et al., 2023c) for tackling the challenges of video generation. For example, the largest U-Net, SD-XL (Podell et al., 2023), has 2.6B parameters, whereas transformers, like PaLM (Narang & Chowdhery, 2022), boast 540B.

Inspired by the above analysis, this study presents a thorough exploration of applying transformers to video diffusion and addresses the unique challenges it poses, such as the accurate capturing of temporal dependencies, the appropriate handling of conditioning information, and unifying diverse video generation tasks. Specifically, we propose Video Diffusion Transformer (VDT) for video generation, which comprises transformer blocks equipped with temporal and spatial attention modules, a VAE tokenizer for effective tokenization, and a decoder to generate video frames. VDT offers several appealing benefits. **1)** It excels at capturing temporal dependencies, including both the evolution of frames and the dynamics of objects over time. The powerful temporal attention module also ensures the generation of high-quality and temporally consistent video frames. **2)** Benefiting from the flexibility and tokenization capabilities of transformers, conditioning the observed video frames is straightforward. For example, a simple token concatenation is sufficient to achieve remarkable performance. **3)** The design of VDT is paired with a unified spatial-temporal mask modeling mechanism, harnessing diverse video generation tasks (see Figure 1), *e.g.*, unconditional video generation, bidirectional video forecasting, arbitrary video interpolation, and dynamic video animation. Our proposed training mechanism positions VDT as a general-purpose video diffuser.

Our contributions are three-fold.

- We pioneer the utilization of transformers in diffusion-based video generation by introducing our Video Diffusion Transformer (VDT). To the best of our knowledge, this marks the first successful model in transformer-based video diffusion, showcasing the potential in this domain.

- We introduce a unified spatial-temporal mask modeling mechanism for VDT, combined with its inherent spatial-temporal modeling capabilities, enabling it to unify a diverse array of general-purpose tasks with state-of-the-art performance, including capturing the dynamics of 3D objects on the physics-QA dataset (Bear et al., 2021).
- We present a comprehensive study on how VDT can capture accurate temporal dependencies, handle conditioning information, and be efficiently trained, etc. By exploring these aspects, we contribute to a deeper understanding of transformer-based video diffusion and advance the field.

## 2 RELATED WORK

**Diffusion Model.** Recently, diffusion models (Sohl-Dickstein et al., 2015; Song & Ermon, 2019; Ho et al., 2020; Choi et al., 2021) have shown great success in the generation field. (Ho et al., 2020) firstly introduced a noise prediction formulation for image generation, which generates images from pure Gaussian noises by denoising noise step by step. Based on such formulation, numerous improvements have been proposed, which mainly focus on sample quality (Rombach et al., 2021), sampling efficiency (Song et al., 2021), and condition generation (Ho & Salimans, 2022). Besides image generation, diffusion models have also been applied to various domains, including audio generation (Kong et al., 2020; Huang et al., 2023), video generation (Ho et al., 2022b), and point cloud generation (Luo & Hu, 2021). Although most of the previous works adopt U-Net based architectures in diffusion model, transformer-based diffusion model has been recently proposed by (Peebles & Xie, 2022; Bao et al., 2022) for image generation, which can achieve comparable results with U-Net based architecture in image generation. In this paper, due to the superior temporal modeling ability of transformer, we explore the use of the transformer-based diffusion model for video generation and prediction.

**Video Generation and Prediction.** Video generation and video prediction are two highly challenging tasks that has gained significant attention in recent years due to the explosive growth of web videos. Previous works (Vondrick et al., 2016; Saito et al., 2017) have adopted GANs to directly learn the joint distribution of video frames, while others (Esser et al., 2021; Ge et al., 2022; Gupta et al., 2023; Yu et al., 2023a) have adopted a vector quantized autoencoder followed by a transformer to learn the distribution in the quantized latent space. For video generation, several poisoners works (Ho et al., 2022b; He et al., 2022; Blattmann et al., 2023; Wang et al., 2023c; Yu et al., 2023b; Wang et al., 2023b) extend the 2D U-Net by incorporating temporal attention into 2D convolution kernels to learn both temporal and spatial features simultaneously. Diffusion has been employed for video prediction tasks in recent works (Voleti et al., 2022; Yang et al., 2022), which utilize specialized modules to incorporate the 2D U-Net network and generate frames based on previously generated frames. Prior research has primarily centered on either video generation or prediction, rarely excelling at both simultaneously. In this paper, we present VDT, a video diffusion model rooted in a pure transformer architecture. Our VDT showcases strong video generation potential and can seamlessly extend to and perform well on a broader array of video generation tasks through our unified spatial-temporal mask modeling mechanism, without requiring modifications to the underlying architecture.

## 3 METHOD

We introduce the Video Diffusion Transformer (VDT) as a unified framework for diffusion-based video generation. We present an overview in Section 3.1, and then delve into the details of applying our VDT to the conditional video generation in Section 3.2. In Section 3.3, we show how VDT be extended for a diverse array of general-purpose tasks via unified spatial-temporal mask modeling.

### 3.1 OVERALL FRAMEWORK

In this paper, we focus on exploring the use of transformer-based diffusion in video generation, and thus adopt the traditional transformer structure for video generation and have not made significant modifications to it. The influence of the transformer architecture in video generation is left to future work. The overall architecture of our proposed video diffusion transformer (VDT) is presented in Fig 2. VDT parameterizes the noise prediction network.

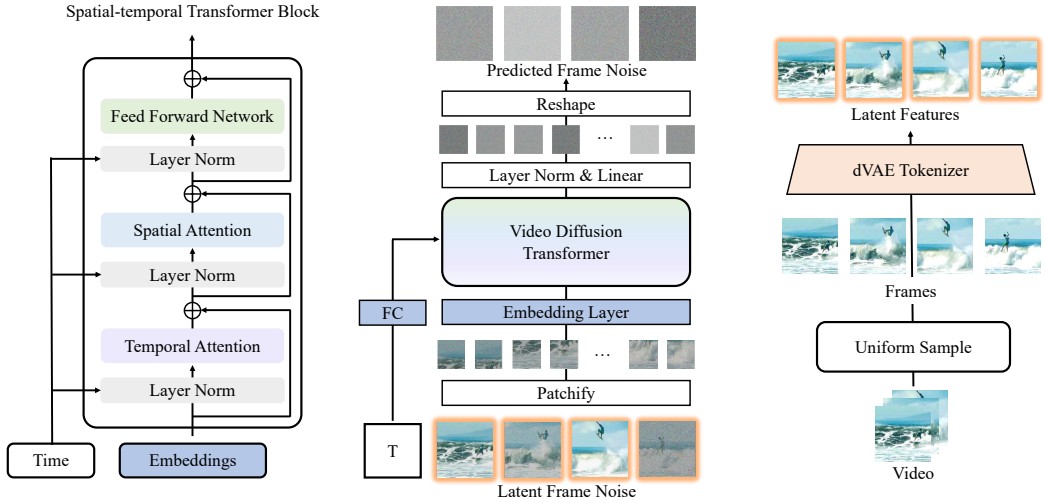

Figure 2: Illustration of our video diffusion transformer (VDT). (a) VDT block with temporal and spatial attention. (b) The diffusion pipeline of our VDT. (c) We uniformly sample frames and then project them into the latent space using a pre-trained VAE tokenizer.

**Input/Output Feature.** The objective of VDT is to generate a video clip $\in R^{F \times H \times W \times 3}$, consisting of $F$ frames of size $H \times W$. However, using raw pixels as input for VDT can lead to extremely heavy computation, particularly when $F$ is large. To address this issue, we take inspiration from the LDM (Rombach et al., 2022) and project the video into a latent space using a pre-trained VAE tokenizer from LDM. This speeds up our VDT by reducing the input and output to latent feature/noise $\mathcal{F} \in R^{F \times H/8 \times W/8 \times C}$, consisting of $F$ frame latent features of size $H/8 \times W/8$. Here, $8$ is the downsample rate of the VAE tokenizer, and $C$ denotes the latent feature dimension.

**Linear Embedding.** Following the approach of Vision Transformer (ViT) (Dosovitskiy et al., 2021), we divide the latent feature representation into non-overlapping patches of size $N \times N$ in the spatial dimension. In order to explicitly learn both spatial and temporal information, we add spatial and temporal positional embeddings (sin-cos) to each patch.

**Spatial-temporal Transformer Block.** Inspired by the success of space-time self-attention in video modeling, we insert a temporal attention layer into the transformer block to obtain the temporal modeling ability. Specifically, each transformer block consists of a multi-head temporal-attention, a multi-head spatial-attention, and a fully connected feed-forward network, as shown in Figure 2.

During the diffusion process, it is essential to incorporate time information into the transformer block. Following the adaptive group normalization used in U-Net based diffusion model, we integrate the time component after the layer normalization in the transformer block, which can be formulated as:

$$adaLN(h, t) = t_{scale} LayerNorm(h) + t_{shift}, \tag{1}$$

where $h$ is the hidden state and $t_{scale}$ and $t_{shift}$ are scale and shift parameters obtained from the time embedding.

## 3.2 CONDITIONAL VIDEO GENERATION SCHEME FOR VIDEO PREDICTION

In this section, we explore how to extend our VDT model to video prediction, or in other words, conditional video generation, where given/observed frames are conditional frames.

**Adaptive layer normalization.** A straightforward approach to achieving video prediction is to incorporate conditional frame features into the layer normalization of transformer block, similar to how we integrate time information into the diffusion process. The Eq 1 can be formulated as:

$$adaLN(h, c) = c_{scale} LayerNorm(h) + c_{shift}, \tag{2}$$

where $h$ is the hidden state and $c_{scale}$ and $c_{shift}$ are scale and shift parameters obtained from the time embedding and condition frames.

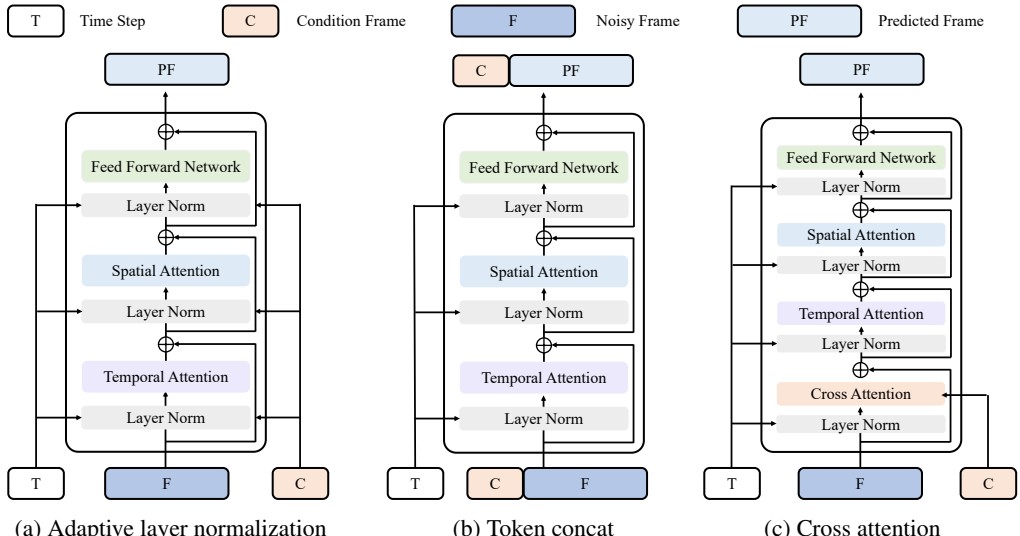

Figure 3: Illustration of three video prediction schemes.

**Cross-attention.** We also explored the use of cross-attention as a video prediction scheme, where the conditional frames are used as keys and values, and the noisy frame serves as the query. This allows for the fusion of conditional information within the noisy frame. Prior to entering the cross-attention layer, the features of the conditional frames are extracted using the VAE tokenizer and being patchfied. Spatial and temporal position embeddings are also added to assist our VDT in learning the corresponding information within the conditional frames.

**Token concatenation.** Our VDT model adopts a pure transformer architecture, therefore, a more intuitive approach is to directly utilize conditional frames as input tokens for VDT. We achieve this by concatenating the conditional frames (latent features) and noisy frames in token level, which is then fed into the VDT. Then we split the output frames sequence from VDT and utilize the predicted frames for the diffusion process, as illustrated in Figure 3 (b). We have found that this scheme exhibits the fastest convergence speed as shown in Figure 6, and compared to the previous two approaches, delivers superior results in the final outcomes.

Furthermore, we discovered that even if we use a fixed length for the conditional frames during the training process, our VDT can still take any larger length of conditional frame as input and output consistent predicted features (more details are provided in Appendix).

### 3.3 UNIFIED SPATIAL-TEMPORAL MASK MODELING

In Section 3.2, we demonstrated that simple token concatenation is sufficient to extend VDT to tasks in video prediction. An intuitive question arises: can we further leverage this scalability to extend VDT to more diverse video generation tasks—such as video frame interpolation—into a single, unified model; without introducing any additional modules or parameters.

Reviewing the functionality of our VDT in both unconditional generation and video prediction, the only difference lies in the type of input features. Specifically, the input can either be pure noise latent features or a concatenation of conditions and noise latent features. Then we introduce a conditional spatial-temporal mask to unified the conditional input $\mathcal{I}$, as formulated in the following equation:

$$\mathcal{I} = \mathcal{F} \wedge (1 - \mathcal{M}) + \mathcal{C} \wedge \mathcal{M}. \tag{3}$$

Here, $\mathcal{C} \in R^{F \times H \times W \times C}$ represents the actual conditional video, $\mathcal{F} \in R^{F \times H \times W \times C}$ signifies noise, $\wedge$ represents bitwise multiplication, and the spatial-temporal mask $\mathcal{M} \in R^{F \times H \times W \times C}$ controls whether each token $t \in R^C$ originates from the real video or noise.

Under this unified framework, we can modulate the the spatial-temporal mask $M$ to incorporate additional video generation tasks into the VDT training process. This ensures that a well-trained VDT can be effortlessly applied to various video generation tasks. Specifically, we consider the following training task during the training (as shown in Figure 4 and Figure 5):

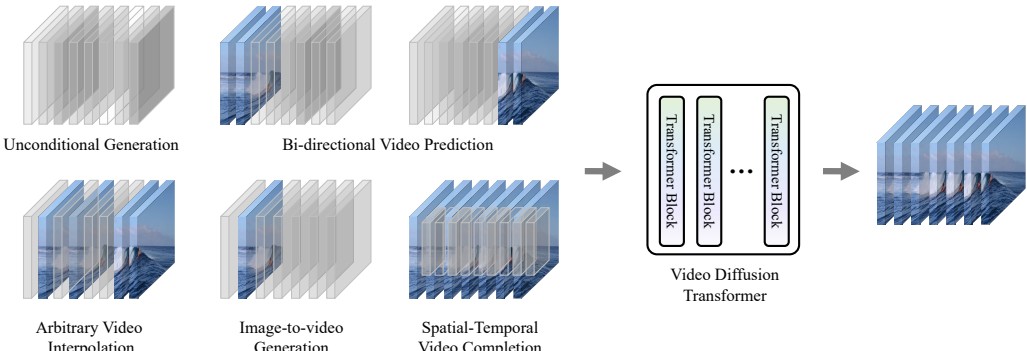

Figure 4: Illustration of our unified spatial-temporal mask modeling mechanism.

**Unconditional Generation** This training task aligns with the procedures outlined in In Section 3.1, where the spatial-temporal $M$ is set to all zero.

**Bi-directional Video Prediction** Building on our extension of VDT to video prediction tasks in Section 3.2, we further augment the complexity of this task. In addition to traditional forward video prediction, we challenge the model to predict past events based on the final frames of a given video, thereby encouraging enhanced temporal modeling capabilities.

**Arbitrary Video Interpolation** Frame interpolation is a pivotal aspect of video generation. Here, we extend this task to cover scenarios where arbitrary n frames are given, and the model is required to fill in the missing frames to complete the entire video sequence.

**Image-to-video Generation** is a specific instance of Arbitrary Video Interpolation. Starting from a single image, we random choose a temporal location and force our VDT to generate the full video. Therefore, during inference, we can arbitrarily specify the image's temporal location and generate a video sequence from it.

**Spatial-Temporal Video Completion** While our previous tasks emphasize temporal modeling, we also delve into extending our model into the spatial domain. With our unified mask modeling mechanism, this is made possible by creating a spatial-temporal mask. However, straightforward random spatial-temporal tasks might be too simple for our VDT since it can easily gather information from surrounding tokens. Drawing inspiration from BEiT (Bao et al., 2021), we adopt a spatial-temporal block mask methodology to preclude the VDT from converging on trivial solutions.

## 4 EXPERIMENT

### 4.1 DATASETS AND SETTINGS.

**Datasets.** The VDT is evaluated on both video generation and video prediction tasks. Unconditional generation results on the widely-used UCF101 (Soomro et al., 2012), TaiChi (Siarohin et al., 2019) and Sky Time-Lapse (Xiong et al., 2018) datasets are provided for video synthesis. For video prediction, experiments are conducted on the real-world driven dataset - Cityscapes (Cordts et al., 2016), as well as on a more challenging physical prediction dataset Physion (Bear et al., 2021) to demonstrate the VDT's strong prediction ability.

**Evaluation.** We adopt Fréchet Video Distance (FVD) (Unterthiner et al., 2018) as the main metric for comparing our model with previous works, as FVD captures both fidelity and diversity of generated samples. Additionally, for video generation tasks, we report the Structural Similarity Index (SSIM) and Peak Signal-to-Noise Ratio (PSNR). VQA accuracy is reported for the physical prediction task. Consistent with previous work (Voleti et al., 2022), we use clip lengths of 16, 30, and 16 for UCF101, Cityscapes, and Physion, respectively. Furthermore, all videos are center-cropped and downsampled to 64x64 for UCF101, 128x128 for Cityscapes and Physion, 256x256 for TaiChi and Sky Time-Lapse.

**VDT configurations.** In Table 1, we provide detailed information about two versions of the VDT model. By default, we utilize VDT-L for all experiments. We empirically set the initial learning rate to 1e-4 and adopt AdamW (Loshchilov & Hutter, 2019) for our training. We utilize a pre-trained

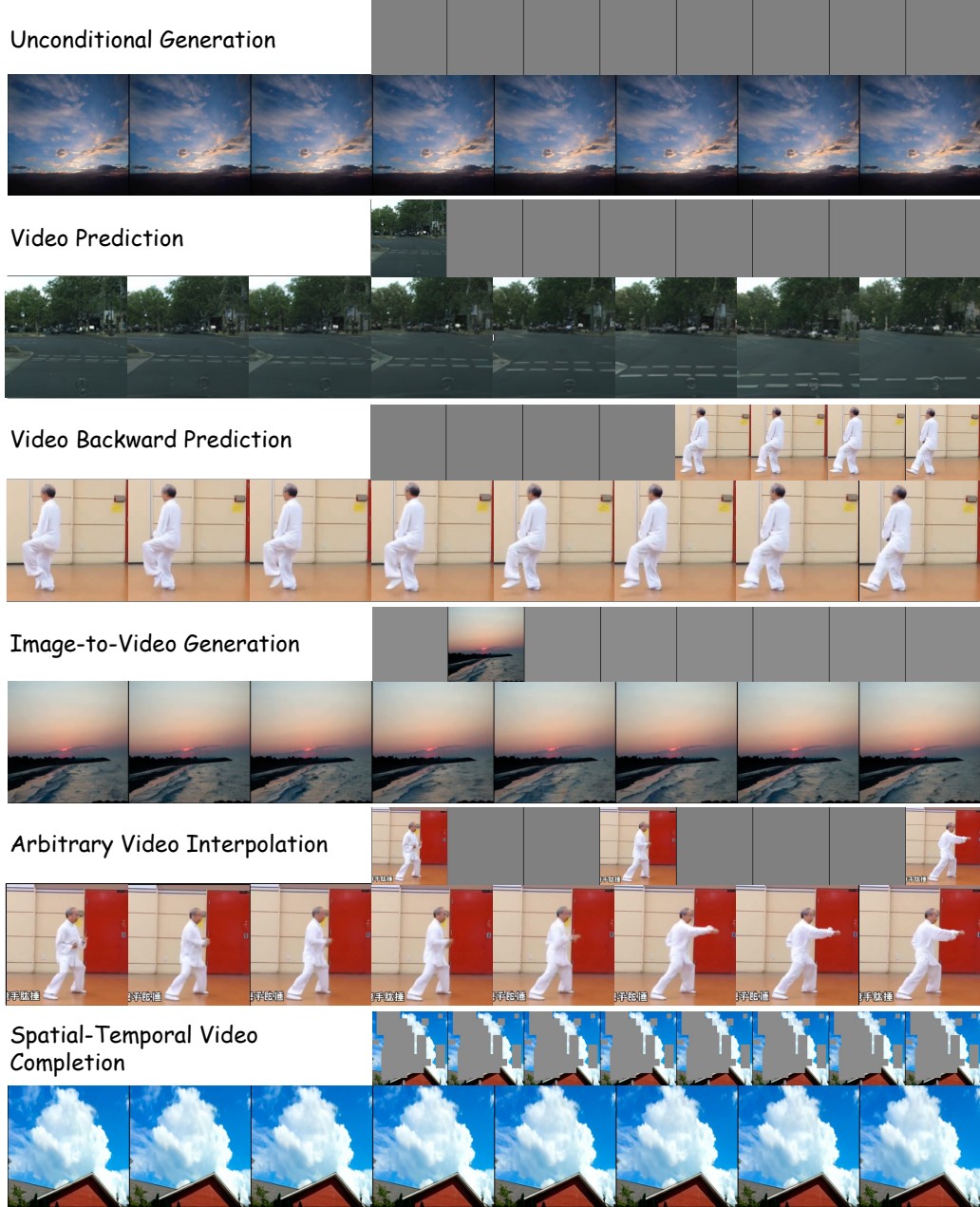

Figure 5: Qualitative results on unified video generation tasks. For each sample, we provide the mask and condition information in the top line, and the result generated by VDT in the bottom.

Table 1: Configurations of VDT. FVD results are reported on UCF101 unconditional generation.

| Model | Layer | Hidden State | Heads | MLP ratio | FVD ↓ |
|-------|-------|--------------|-------|-----------|-------|
| VDT-S | 12 | 384 | 6 | 4 | 425.6 |
| VDT-L | 28 | 1152 | 16 | 4 | 225.7 |

variational autoencoder (VAE) model (Rombach et al., 2022) as the tokenizer and freeze it during training. The hyper-parameters are uniformly set to Patchsize = 2. More details are given in Appendix.

## 4.2 ANALYSIS

**Different conditional strategy for video prediction.** In Section 3.2, we explore three conditional strategies: (1) adaptive layer normalization, (2) cross-attention, and (3) token concatenation. The

Table 2: Video prediction on Physion (128 × 128) conditioning on 8 frames and predicting 8. We compare three video prediction schemes.

| Methods | FVD ↓ | SSIM ↑ | PSNR ↑ |
|---|---|---|---|
| Ada. LN | 270.8 | 0.6247 | 16.8 |
| Cross-Attention | 134.9 | 0.8523 | 28.6 |
| Token Concat | 129.1 | 0.8718 | 30.2 |

Table 3: Different training strategies of VDT-S on UCF101. S: spatial train only, J: joint train.

| Method | S | T | FVD↓ | Time |
|---|---|---|---|---|
| J directly | 0 | 40k | 554.8 | 7.2 |
| J directly | 0 | 80k | 451.9 | 14.4 |
| J directly | 0 | 120k | 425.6 | 21.5 |
| S pre. then J | 80k | 40k | 431.7 | 11.2 |

Table 4: Unconditional video generation results on UCF-101. * means trained on full split (train + test).

| Method | Resolution | FVD ↓ |
|---|---|---|
| **GAN:** | | |
| TGANv2 (Saito et al., 2020) | 16×128×128 | 1209.0 |
| MoCoGAN* (Tulyakov et al., 2018) | 16×128×128 | 838.0 |
| DIGAN* (Yu et al., 2022) | 16×128×128 | 577.0 |
| **Diff. based on U-Net, Large Pre:** | | |
| Latent-Shift* (An et al., 2023) | 16×256×256 | 360.0 |
| VideoGen* (Li et al., 2023) | 16×256×256 | 345.0 |
| VideoFusion* (Luo et al., 2023) | 16×128×128 | 220.0 |
| Make-A-Video* (Singer et al., 2022) | 16×256×256 | 81.3 |
| **Diff. based on U-Net:** | | |
| PVDM* (Yu et al., 2023b) | 16×256×256 | 343.6 |
| MCVD (Voleti et al., 2022) | 16×64×64 | 1143.0 |
| PYoCo* (Ge et al., 2023) | 16×64×64 | 310.0 |
| VDM* (Ho et al., 2022b) | 16×64×64 | 295.0 |
| **Diff. based on Transformer:** | | |
| VDT | 16×64×64 | **225.7** |

Figure 6: Training loss on three video prediction schemes. Token concatenation approach achieved the fastest convergence speed and the lowest loss.

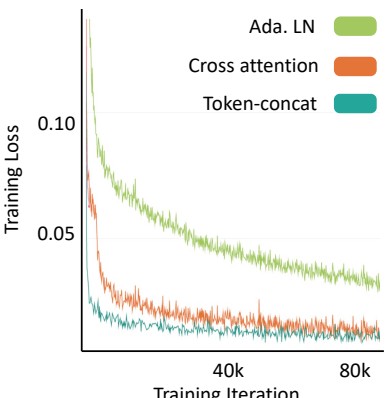

results of convergence speed and prediction performance are presented in Figure 3 and Table 2, respectively. Notably, the token concatenation strategy achieves the fastest convergence speed and the best prediction performance (i.e., FVD and SSIM in Table 2). As a result, we adopt the token concatenation strategy for all video prediction tasks in this paper.

**Training strategy.** In this part, we investigate different training strategies in Table 3. For spatial-only training, we remove the temporal attention in each block and sample one frame from each video to force the model to learn the spatial information. This enables the model to focus on learning spatial features separately from temporal features. It is evident that that spatial pretraining then joint training outperforms directly spatial-temporal joint tuning (431.7 vs. 451.9) with significantly less time (11.2 vs. 14.4), indicating the crucial role of image pretraining initialization in video generation.

### 4.3 COMPARISON TO THE STATE-OF-THE-ARTS

**Unconditional Generation.** The quantitative results in unconditional generation are given in Table 4. Our VDT demonstrates significant superiority over all GAN-based methods. Although MCVD (Voleti et al., 2022) falls under the diffusion-based category, our VDT outperforms it by a significant margin. This difference in performance may be attributed to the fact that MCVD is specifically designed for video prediction tasks. VDM (Ho et al., 2022b) is the most closely related method, as it employs a 2D U-Net with additional temporal attention. However, direct comparisons are not feasible as VDM only presents results on the train+test split. Nevertheless, our VDT achieves superior performance, even with training solely on the train split.

We also conducted a qualitative analysis in Figure 7, focusing on TaiChi (Siarohin et al., 2019) and Sky Time-Lapse (Xiong et al., 2018). It is evident that both DIGAN and VideoFusion exhibit noise artifacts in the Sky scene, whereas our VDT model achieves superior color fidelity. In the TaiChi, DIGAN and VideoFusion predominantly produce static character movements, accompanied by distortions in the hand region. Conversely, our VDT model demonstrates the ability to generate coherent and extensive motion patterns while preserving intricate details.

Table 5: Video prediction on Cityscapes (128 × 128) conditioning on 2 and predicting 28 frames.

| Cityscapes | FVD↓ | SSIM↑ |
|---|---|---|
| SVG-LP (Denton & Fergus, 2018) | 1300.3 | 0.574 |
| vRNN 1L (Castrejon et al., 2019) | 682.1 | 0.609 |
| Hier-vRNN (Castrejon et al., 2019) | 567.5 | 0.628 |
| GHVAE (Wu et al., 2021) | 418.0 | 0.740 |
| MCVD-spatin (Voleti et al., 2022) | 184.8 | 0.720 |
| MCVD-concat (Voleti et al., 2022) | **141.4** | 0.690 |
| VDT | 142.3 | **0.880** |

Table 6: VQA accuracy on Physion-Collide.

| Model | Accuracy |
|---|---|
| **Object centric:** | |
| Human (upper bound) | 80.0 |
| SlotFormer (Wu et al., 2022b) | 69.3 |
| **Scene centric:** | |
| PRIN (Qi et al., 2021) | 57.9 |
| pVGG-lstm (Bear et al., 2021) | 58.7 |
| pDEIT-lstm (Bear et al., 2021) | 63.1 |
| **VDT (Ours)** | **65.3** |

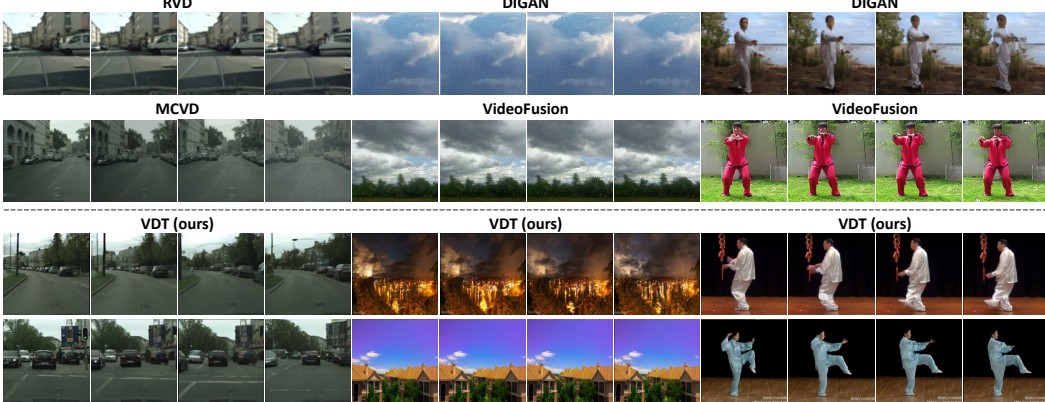

Figure 7: Qualitative video results on video prediction tasks (Cityscapes, 128×128) and video generation tasks (TaiChi-HD and Sky Time-Lapse, 256×256).

**Video Prediction.** Video Prediction is another crucial task in video diffusion. Different from previous works (Voleti et al., 2022) specially designing a diffusion-based architecture to adopt 2D U-Net in video prediction task, the inherent sequence modeling capability of transformers allows our VDT for seamless extension to video prediction tasks. We evaluate it on the Cityscape dataset in Table 5 and Figure 7. It can be observed that our VDT is comparable to MCVD (Voleti et al., 2022) in terms of FVD and superior in terms of SSIM, although we employ a straightforward token concatenation strategy. Additionally, we observe that existing prediction methods often suffer from brightness and color shifts during the prediction process as shown in Figure 7. However, our VDT maintains remarkable overall color consistency in the generated videos. These findings demonstrate the impressive video prediction capabilities of VDT.

**Physical Video Prediction.** We further evaluate our VDT model on the highly challenging Physion dataset. Physion is a physical prediction dataset, specifically designed to assess a model's capability to forecast the temporal evolution of physical scenarios. In contrast to previous object-centric approaches that involve extracting objects and subsequently modeling the physical processes, our VDT tackles the video prediction task directly. It effectively learns the underlying physical phenomena within the conditional frames while generating accurate video predictions. We conducted a VQA test following the official approach, as shown in Table 6. In this test, a simple MLP is applied to the observed frames and the predicted frames to determine whether two objects collide. Our VDT model outperforms all scene-centric methods in this task. These results provide strong evidence of the impressive physical video prediction capabilities of our VDT model.

## 5 CONCLUSION

In this paper, we introduce the Video Diffusion Transformer (VDT), a video generation model based on a simple yet effective transformer architecture. The inherent sequence modeling capability of transformers allows for seamless extension to video prediction tasks using a straightforward token concatenation strategy. Our experimental evaluation, both quantitatively and qualitatively, demonstrates the remarkable potential of the VDT in advancing the field of video generation. We believe our work will serve as an inspiration for future research in the field of video generation.

ACKNOWLEDGEMENTS

This work was supported by National Natural Science Foundation of China (62376274).

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

Table 7: Ablation study on patch size.

| Patch | GFlops | FVD |
|---|---|---|
| 4 | 1.9 | 643.9 |
| 2 | 7.7 | 554.8 |
| 1 | 30.9 | 466.2 |

Table 8: Ablation study on attention head.

| Head | GFlops | FVD |
|---|---|---|
| 3 | 7.7 | 559.8 |
| 6 | 7.7 | 554.8 |
| 12 | 7.7 | 598.7 |

Table 9: Ablation study on layer.

| Layer | GFlops | FVD |
|---|---|---|
| 6 | 3.9 | 580.7 |
| 12 | 7.7 | 554.8 |
| 18 | 11.6 | 500.6 |

Table 10: Ablation study on hidden size.

| Hidden Size | GFlops | FVD |
|---|---|---|
| 192 | 1.9 | 704.2 |
| 384 | 7.7 | 554.8 |
| 768 | 30.9 | 464.2 |

Table 11: Ablation study on architecture.

| Architecture | GFlops | FVD |
|---|---|---|
| Spatial First | 7.7 | 550.2 |
| Temporal First | 7.7 | 554.8 |

**Limitation and broader impacts.** Due to the limitations of our GPU computing resources, we were unable to pretrain our VDT model on large-scale image or video datasets, which restricts its potential. In future research, we aim to address this limitation by conducting pretraining on larger datasets. Furthermore, we plan to explore the incorporation of other modalities, such as text, into our VDT model. For video generation, it is essential to conduct a thorough analysis of the potential consequences and adopt responsible practices to address any negative impacts.

## A  ADDITIONAL ABLATION STUDY

In this part, we conduct a detailed ablation study on UCF101 with VDT-S in Table 7, 8, 9, 10, 11. The results show that reducing the Patchsize, increasing the number of Layers, and increasing the Hidden Size all further improve the model's performance. The position of Temporal and Spatial attention and the number of attention heads do not significantly affect the model's results. Further comprehensive analysis indicates that generally, **an increase in GFlops leads to better results, demonstrating the scalability of VDT**. When maintaining the same GFlops, some trade-offs in design are necessary, but overall, the model's performance does not differ significantly.

## B  TRAINING AND INFERENCE COST

We list training and inference times in Table 12, all experiments are conducted on NVIDIA A100 GPUs.

## C  DETAILS OF DOWNSTREAM TASKS

We list hyperparameters and training details for downstream tasks in Table 13.

## D  PHYSICAL VIDEO PREDICTION.

Most video prediction task was designed based on a limited number of short frames to predict the subsequent video sequence. However, in many complex real-world scenarios, the conditioning information can be highly intricate and cannot be adequately summarized by just a few frames. As a result, it becomes crucial for the model to possess a comprehensive understanding of the conditioning information in order to accurately generate prediction frames while maintaining semantic coherence.

Table 12: Training and inference times (per sample).

|        | Resolution | VAE    | Training | Inference (t=256) |
|--------|-----------|--------|----------|-------------------|
| VDT-S  | 64x64     | 0.0042 | 0.022    | 1.10              |
| VDT-L  | 64x64     | 0.0042 | 0.051    | 2.57              |
| VDT-S  | 128x128   | 0.0051 | 0.024    | 1.21              |
| VDT-L  | 128x128   | 0.0051 | 0.057    | 6.3               |
| VDT-S  | 256x256   | 0.0058 | 0.026    | 2.63              |
| VDT-L  | 256x256   | 0.0058 | 0.111    | 25.31             |

**VideoFusion**

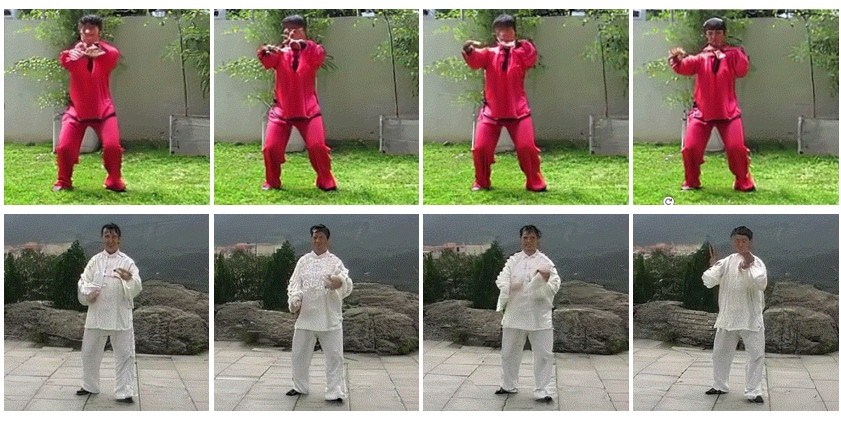

**VDT (ours)**

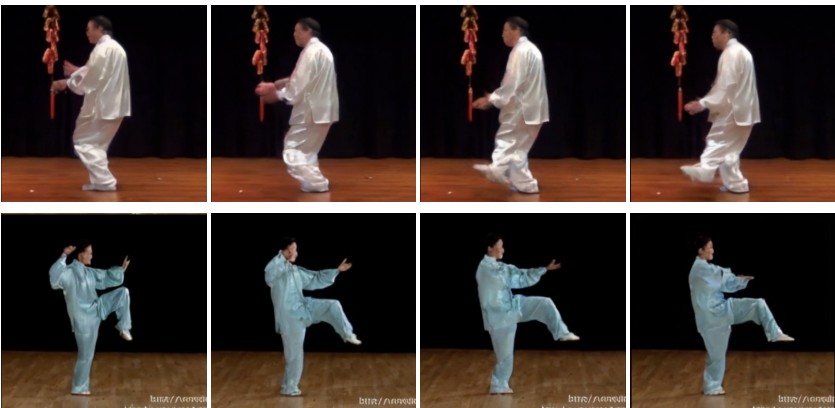

Figure 8:  Qualitative results on TaiChi-HD.

Therefore, we further evaluate our VDT model on the highly challenging Physion dataset. Physion is a physical prediction dataset, specifically designed to assess a model's capability to forecast the temporal evolution of physical scenarios. It offers a more comprehensive and demanding benchmark compared to previous datasets. In contrast to previous object-centric approaches that involve extracting objects and subsequently modeling the physical processes, our VDT tackles the video prediction task directly. It effectively learns the underlying physical phenomena within the conditional frames while generating accurate video predictions.

Specifically, we uniformly sample 8 frames from the observed set of each video as conditional frames and predict the subsequent 8 frames for physical prediction. We present qualitative results in Figure 17 to showcase the quality of our predictions. Our VDT exhibits a strong understanding of the underlying physical processes in different samples, which demonstrates a comprehensive understanding of conditional physical information. Meanwhile, our VDT maintains a high level of

Table 13: Hyperparameters for each task.

| Config | Unconditional Generation | | | Video Prediction | |
|---|---|---|---|---|---|
| | UCF101 | Sky Time-Lapse | TaiChi | CityScapes | Physion |
| optimizer | AdamW | AdamW | AdamW | AdamW | AdamW |
| weight decay | 0 | 0 | 0 | 0 | 0 |
| learning rate | 1e-4 | 1e-4 | 1e-4 | 1e-4 | 1e-4 |
| diffusion noise schedule | linear | linear | linear | linear | linear |
| EMA | 0.9999 | 0.9999 | 0.9999 | 0.9999 | 0.9999 |
| batch size (stage 1) | 256 | 256 | 256 | 128 | 128 |
| batch size (stage 2) | 128 | 64 | 64 | 32 | 32 |
| training resolution | 16x64x64 | 16x256x256 | 16x256x256 | 16x128x128 | 16x128x128 |
| inference resolution | 16x64x64 | 16x256x256 | 16x256x256 | 16x128x128 | 16x128x128 |
| frameskip | 1 | 1 | 1 | 1 | 3 |

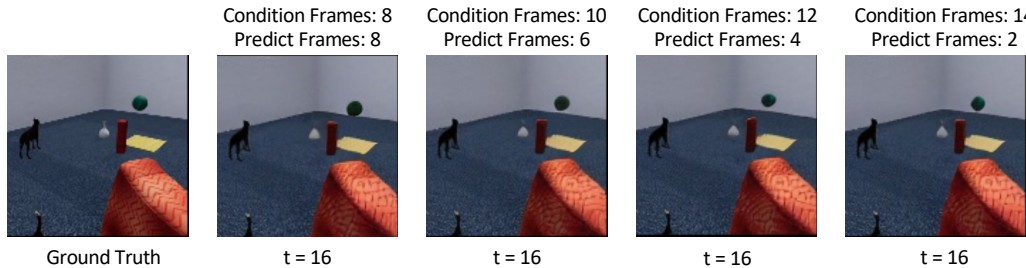

Figure 9: Video prediction results (16x128x128) of frame 16th on the Physion dataset (Bear et al., 2021). During training, we utilize 8 frames as conditional frames and predict the subsequent 8 frames. Then we zero-shot transfer our VDT to condition frames of different larger sizes during inference. We observe that our VDT can perfectly generalize to downstream tasks of different lengths without any additional training.

semantic consistency. Furthermore, we also conducted a VQA test following the official approach, as shown in Table 6. In this test, a simple MLP is applied to the observed frames and the predicted frames to determine whether two objects collide. Our VDT model outperforms all scene-centric methods in this task. These results provide strong evidence of the impressive physical video prediction capabilities of our VDT model.

# E  ZERO-SHOT ADAPTATION TO LONGER CONDITIONAL FRAMES

In our experiment, we find that despite training our VDT (Variable Duration Transformer) with fixed-length condition frames, during the inference process, our VDT can zero-shot transfer to condition frames of different sizes. We illustrate this example in Figure 19 and Figure 9. In training, the condition frames were set to a fixed length of 8. However, during inference, we selected condition frames of lengths 8, 10, 12, and 14, and we observed that the model could perfectly generalize to downstream tasks of different lengths without any additional training. Moreover, the model naturally learned additional information from the extended condition frames. As shown in Figure 9, the prediction of the sample with conditional frame length 14 is more accurate at the 16th frame compared to the sample with conditional frame length 8.

# F  MORE QUALITATIVE RESULTS

We provide more qualitative results in Figure 10, 11, 12, 13, 14 , 15, 16, 17, 18, 19, 18, and 21.

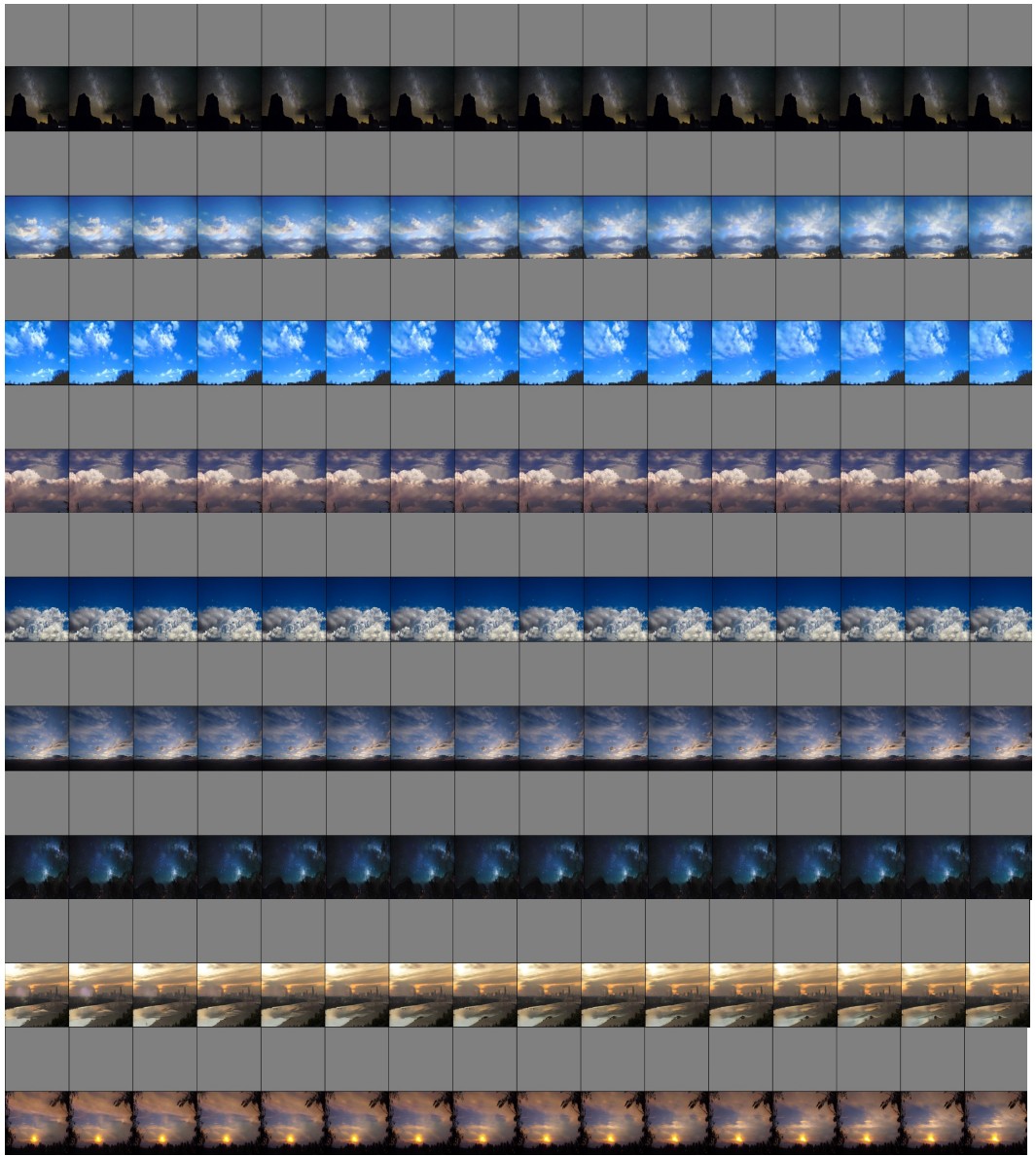

Figure 10: Qualitative results (16x256x256) on Sky Time-Lapse.

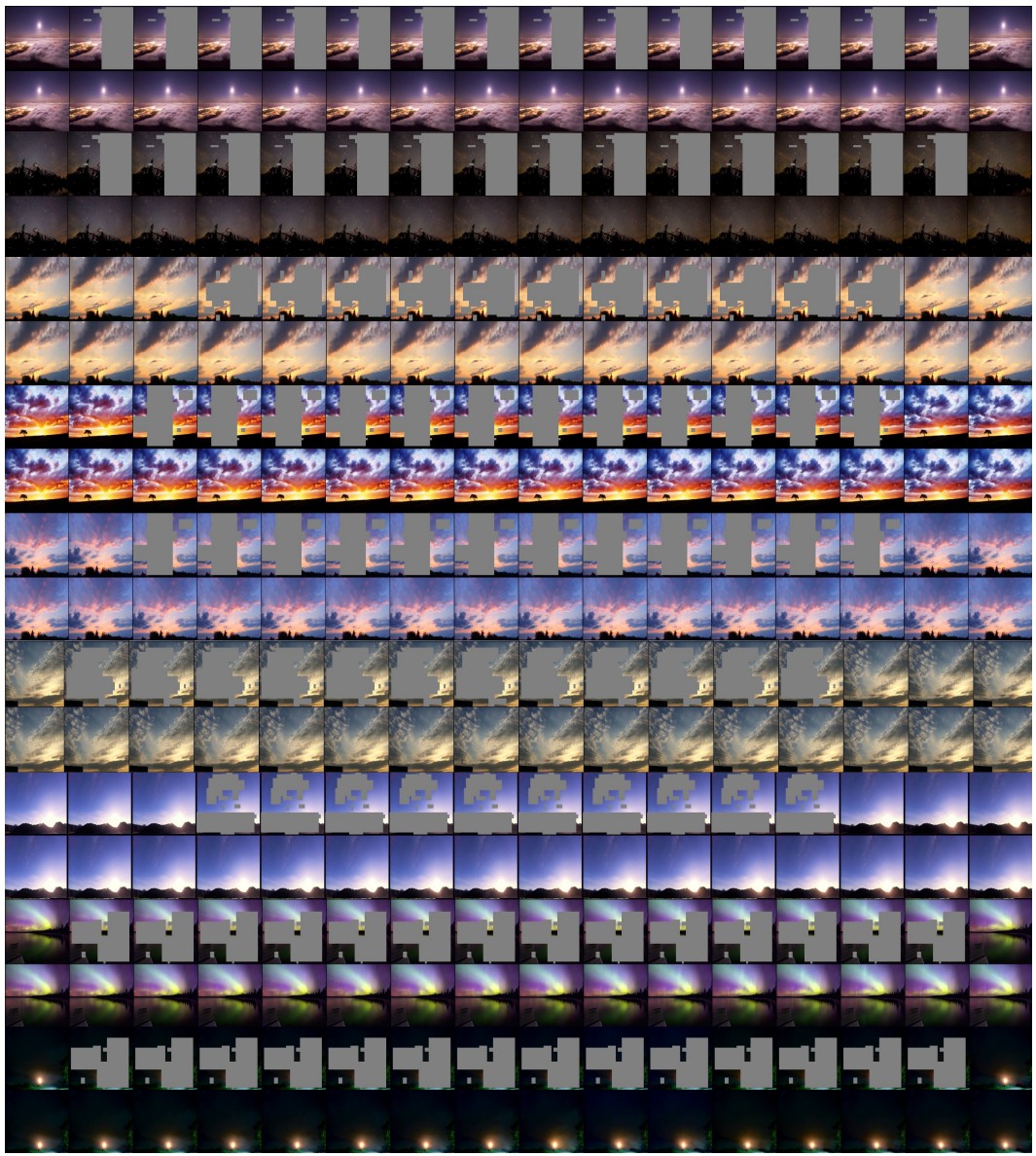

Figure 11: Qualitative results (16x256x256) on Sky Time-Lapse.

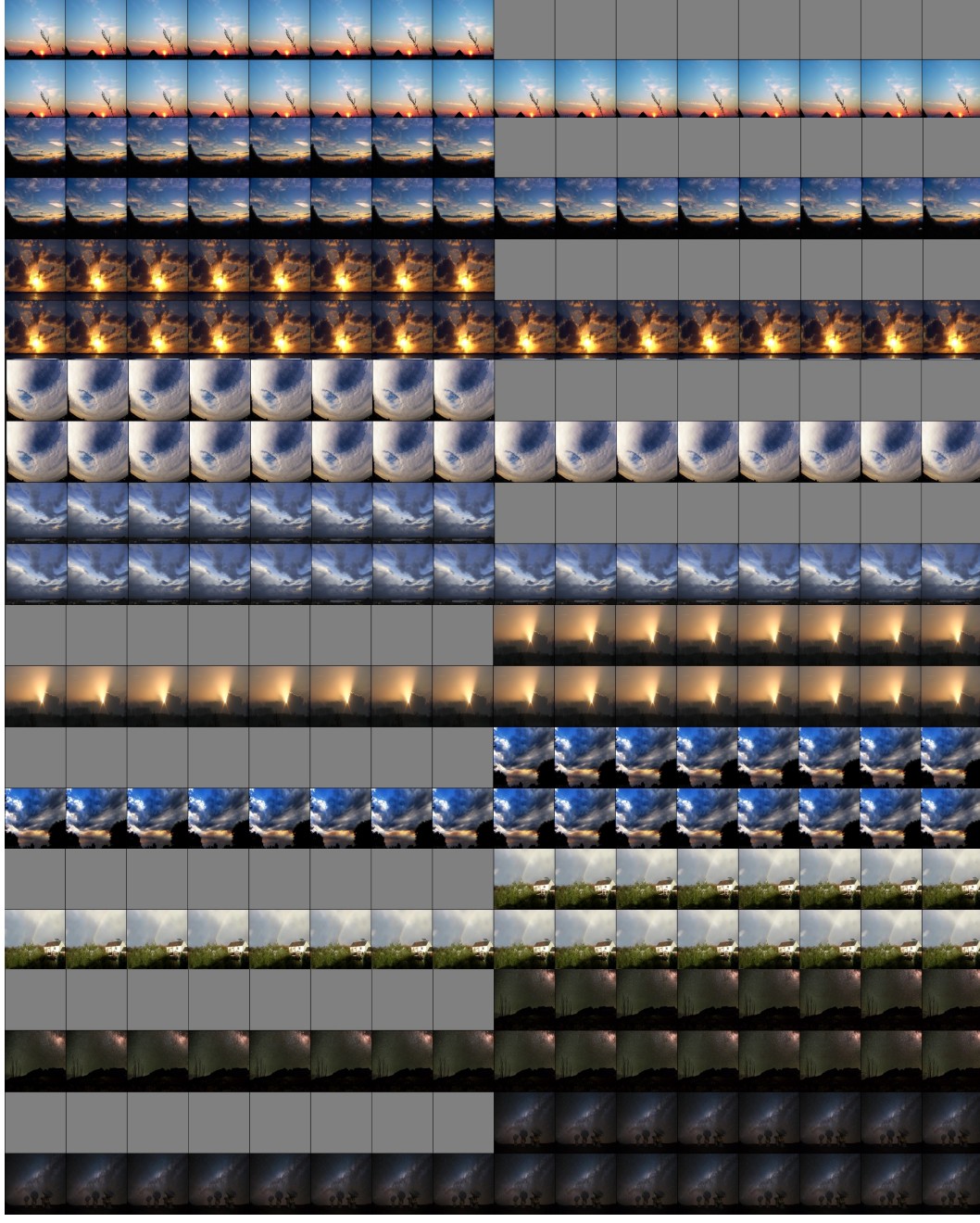

Figure 12: Qualitative results (16x256x256) on Sky Time-Lapse.

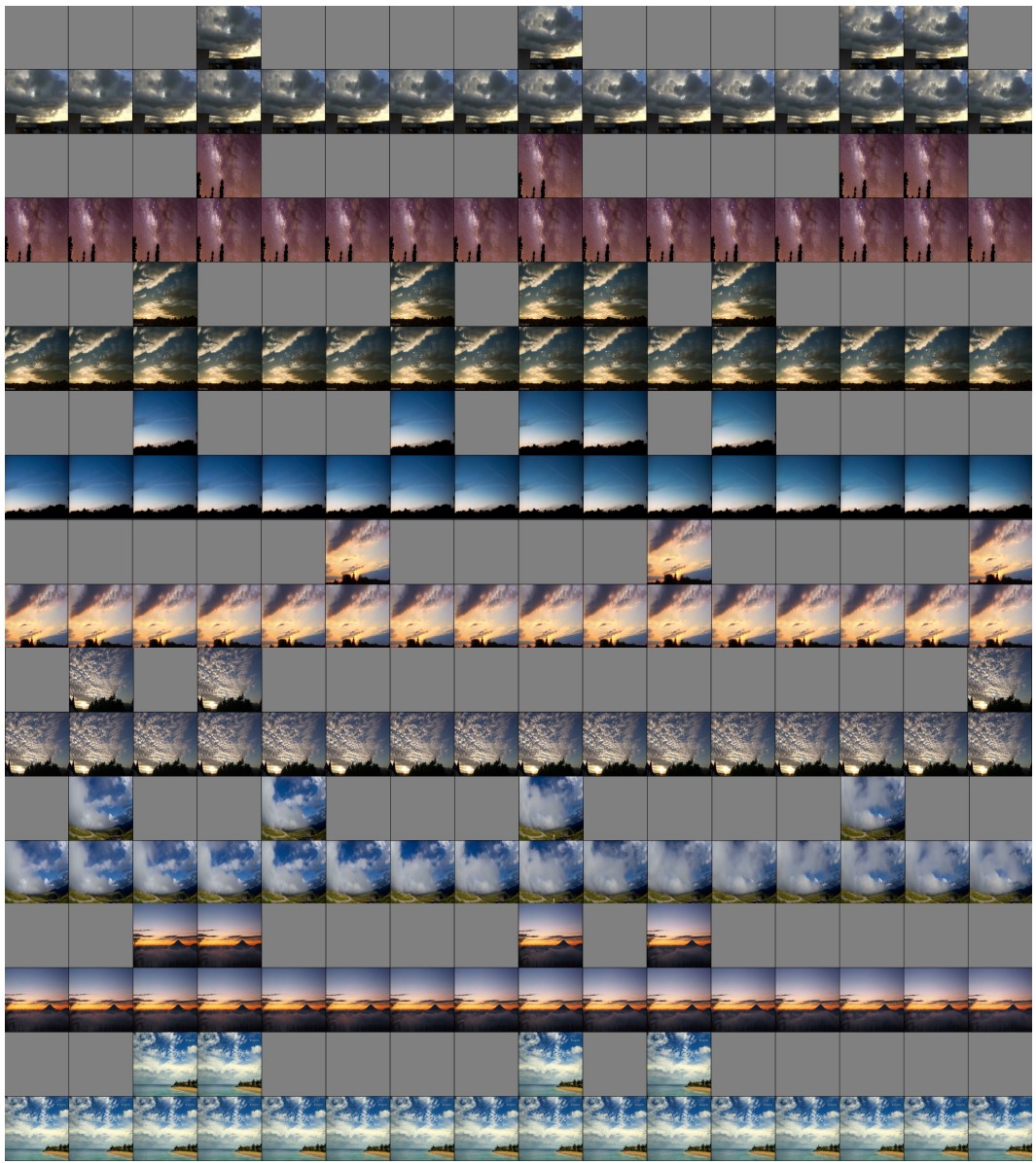

Figure 13: Qualitative results (16x256x256) on Sky Time-Lapse.

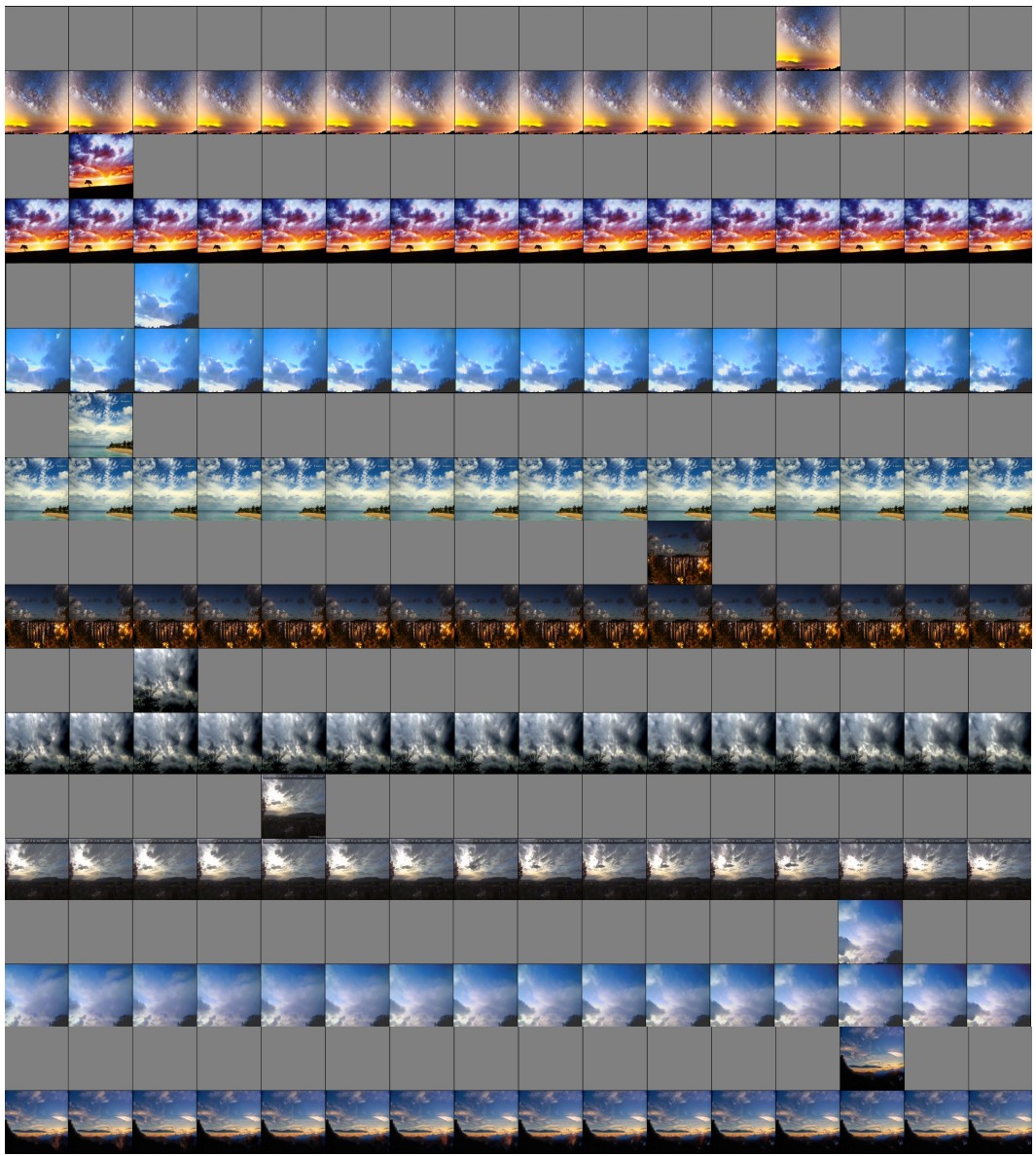

Figure 14: Qualitative results (16x256x256) on Sky Time-Lapse.

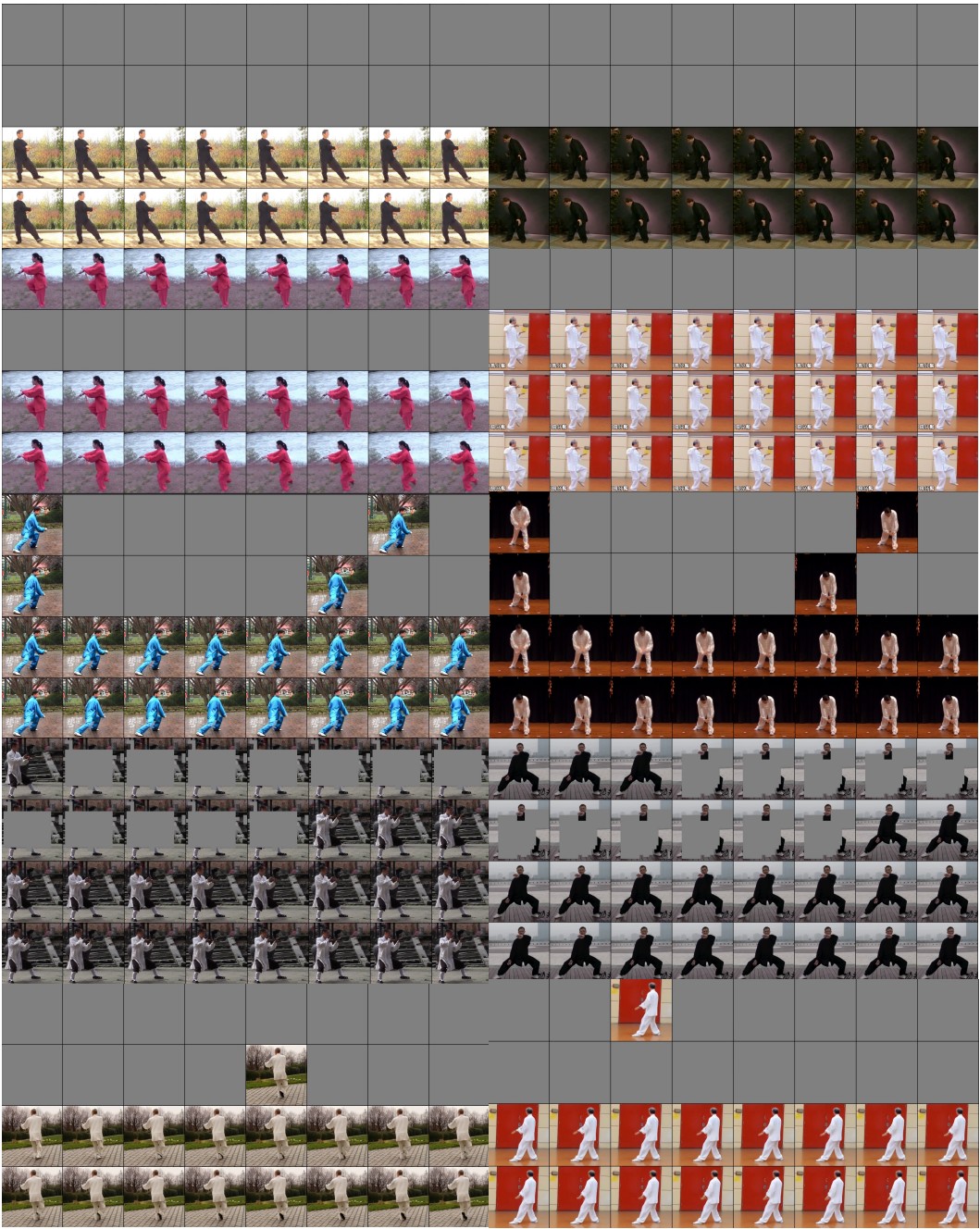

Figure 15: Qualitative results (16x256x256) on TaiChi-HD.

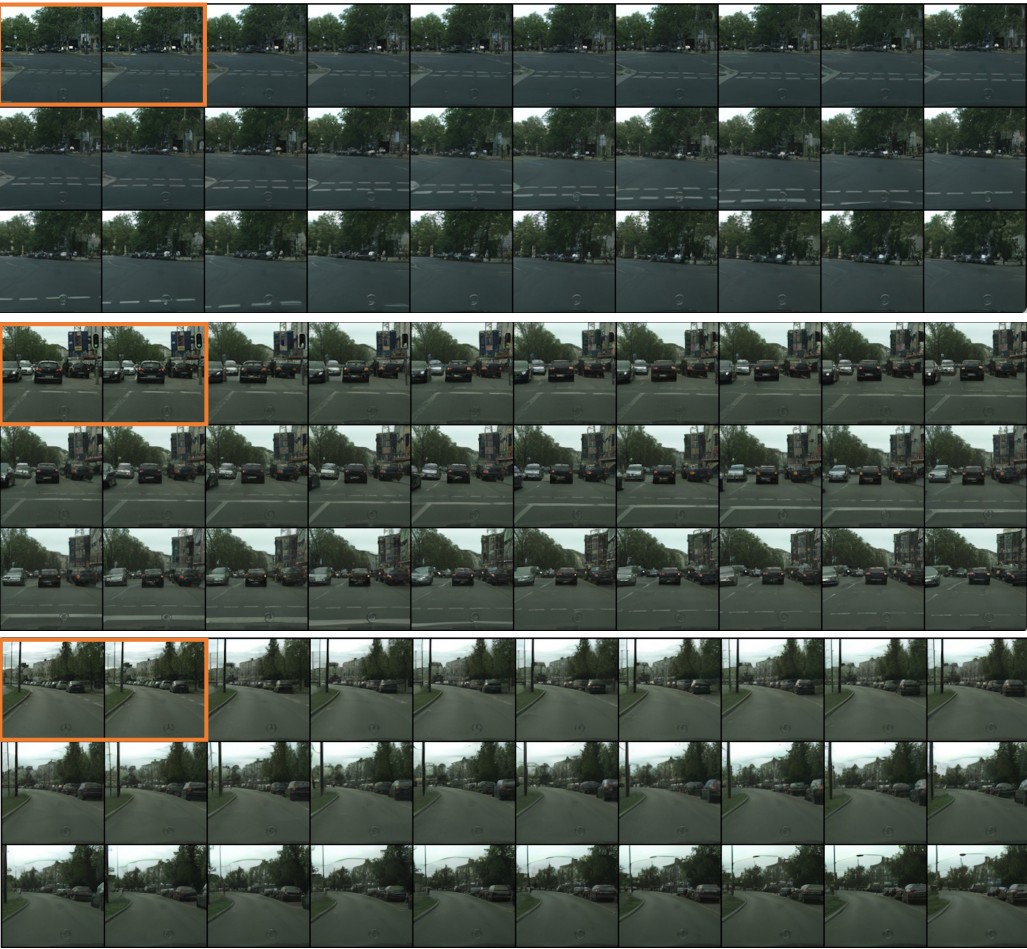

Figure 16: Qualitative video prediction results on Cityscapes (16x128x128), where we utilize 2 frames as conditional frames and predict the subsequent 28 frames in a single forward pass. The predicted frames exhibit semantic coherence, maintaining a high level of consistency in terms of color and brightness.

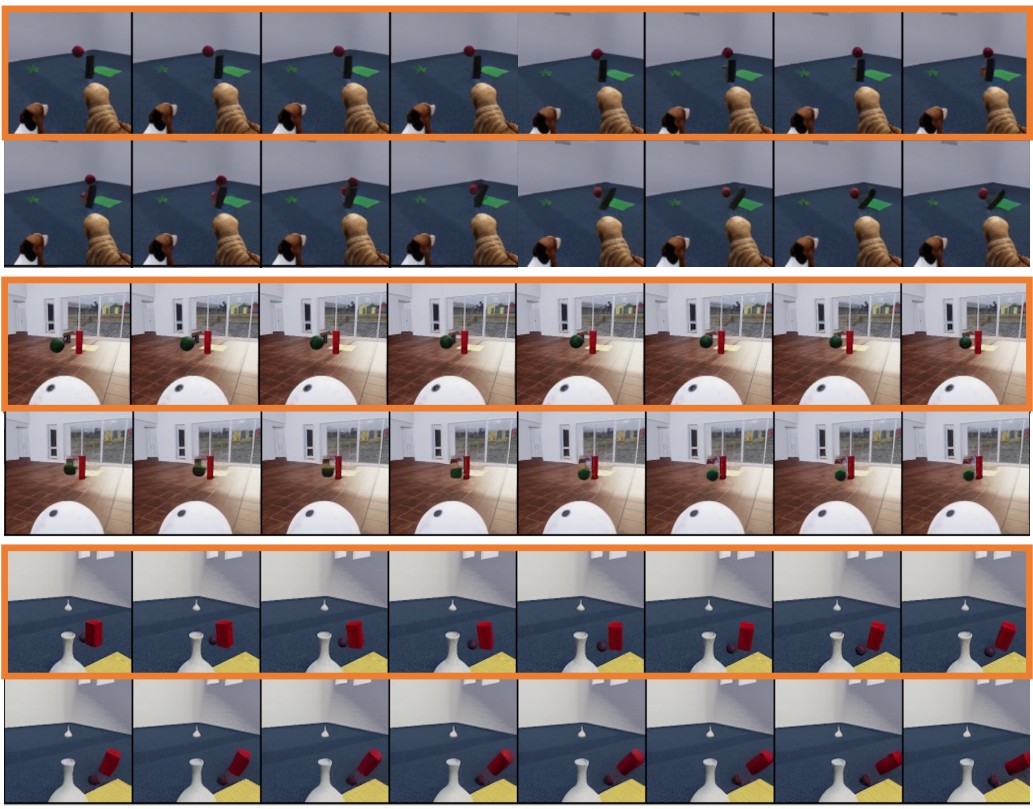

Figure 17: Qualitative video prediction results on the Physion dataset (Bear et al., 2021), where we utilize 8 frames as conditional frames and predict the subsequent 8 frames. In the first example (top two rows) and the third example (bottom two rows), the VDT successfully simulates the physical processes of a ball following a parabolic trajectory and a ball rolling on a flat surface and colliding with a cylinder. In the second example (middle two rows), the VDT captures the velocity/momentum of the ball, as the ball comes to a stop before colliding with the cylinder.

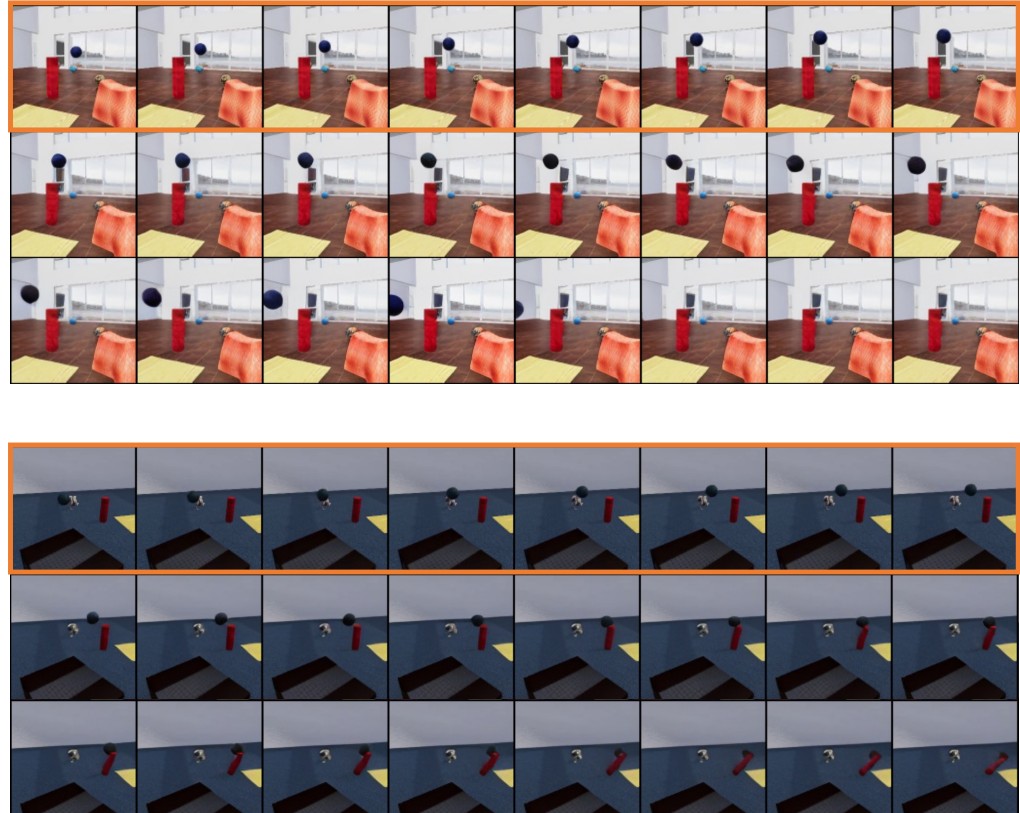

Figure 18: Longer video prediction results (16x128x128) on the Physion dataset (Bear et al., 2021), where we utilize 8 frames as conditional frames and predict the following 8 frames. Subsequently, we predict the next 8 frames based on the previously predicted frames, resulting in a total prediction of 16 frames.

Ground Truth

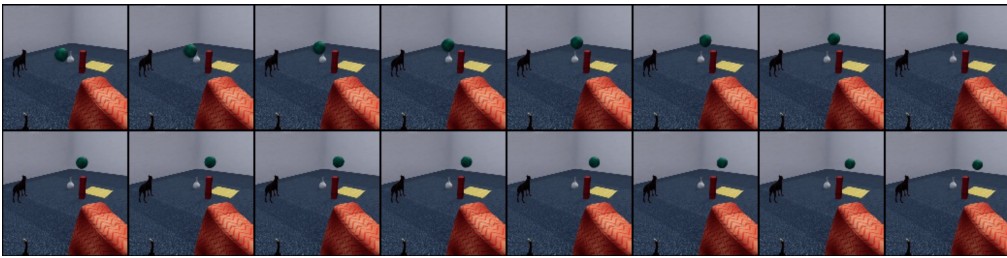

Condition Frame Length: 8  Predict Frame Length: 8

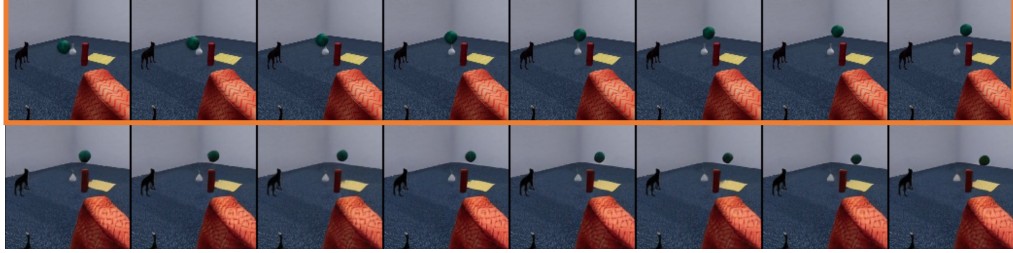

Condition Frame Length: 10 Predict Frame Length: 6

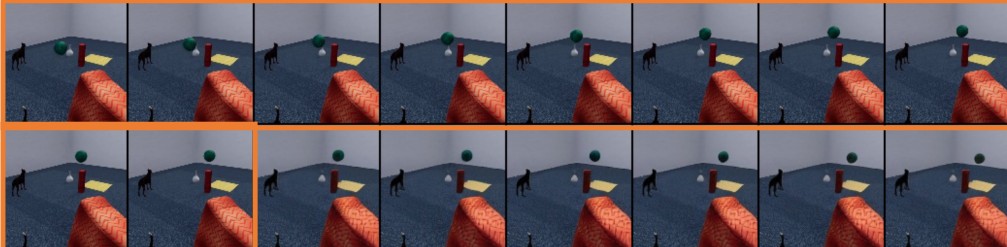

Condition Frame Length: 12 Predict Frame Length: 4

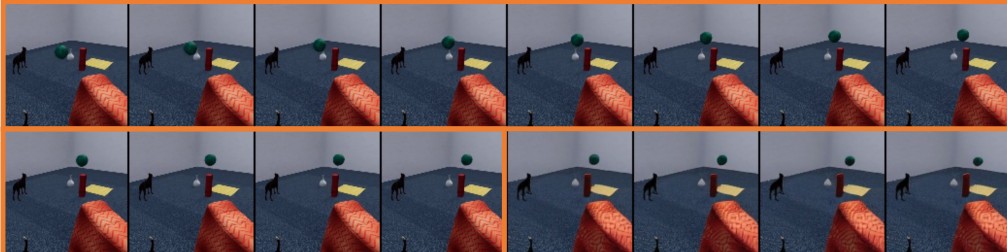

Condition Frame Length: 14 Predict Frame Length: 2

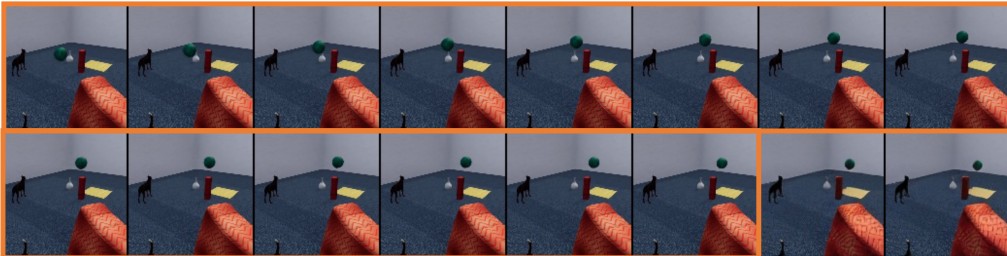

Figure 19: Qualitative video prediction results (16x128x128) on the Physion dataset (Bear et al., 2021). During training, we utilize 8 frames as conditional frames and predict the subsequent 8 frames. Then we zero-shot transfer our VDT to condition frames of different sizes during inference. We observe that our VDT can perfectly generalize to downstream tasks of different lengths without any additional training.

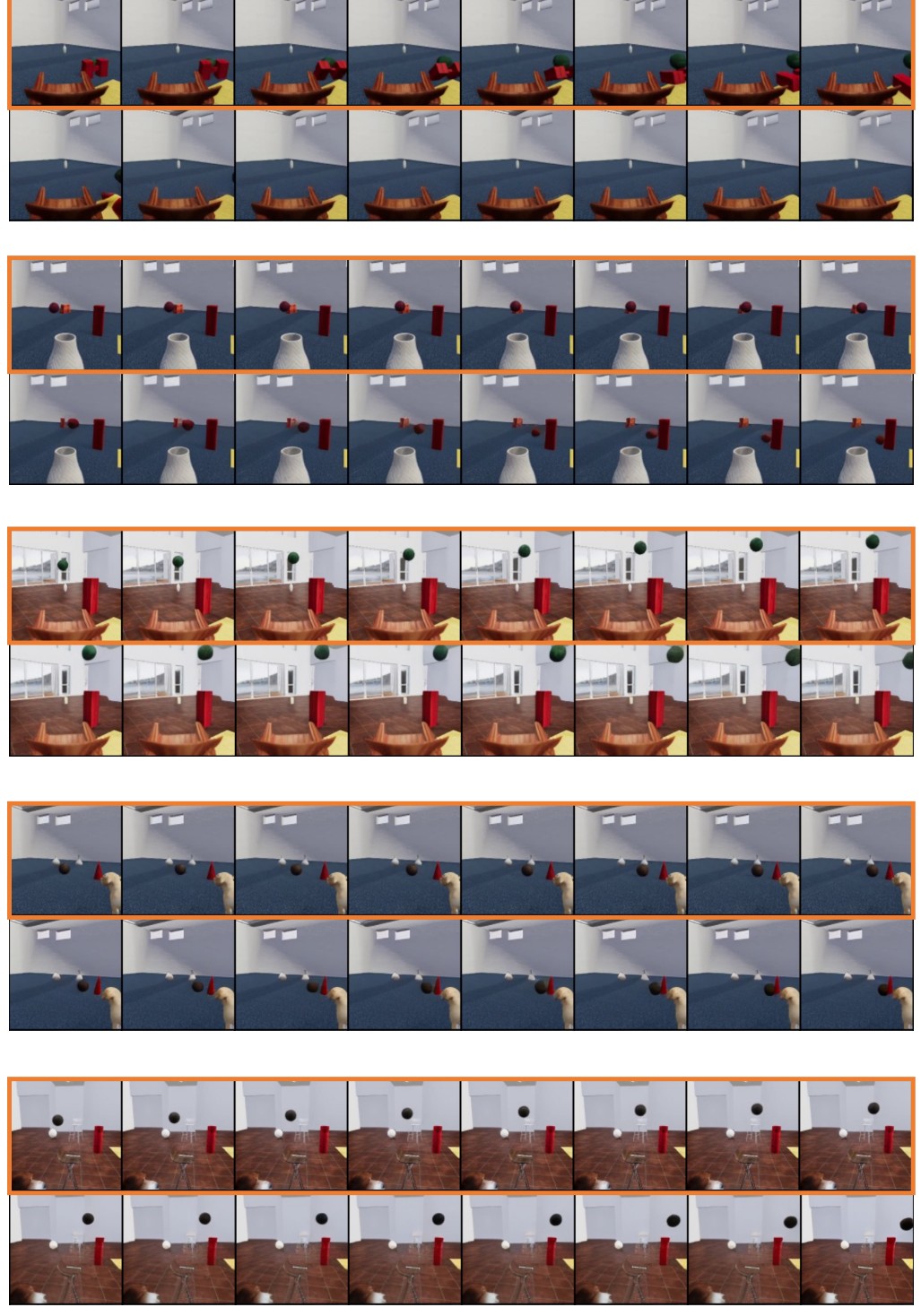

Figure 20: Qualitative video prediction results (16x128x128) on the Physion dataset (Bear et al., 2021), where we utilize 8 frames as conditional frames and predict the subsequent 8 frames.

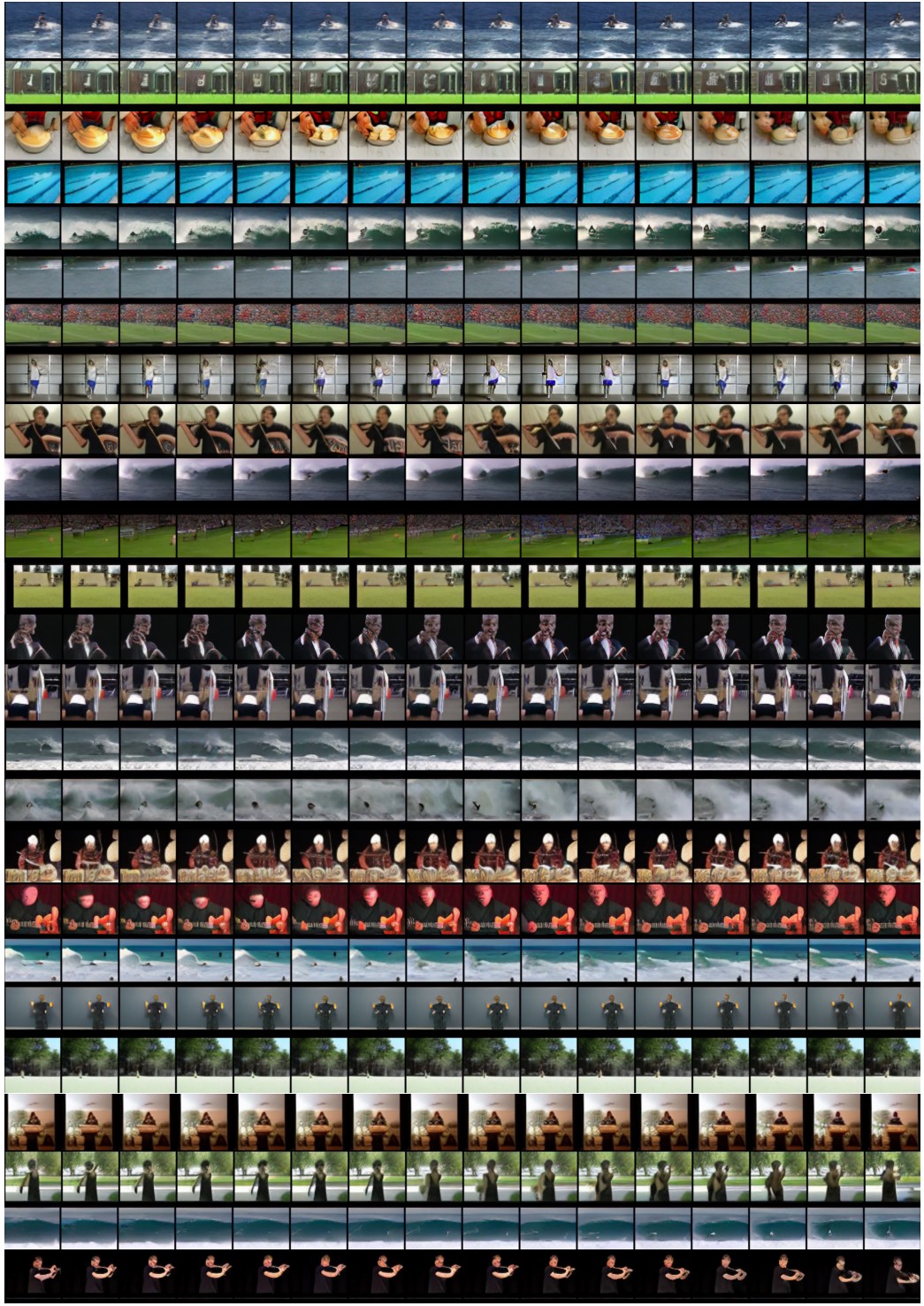

Figure 21: Qualitative unconditional video generation results (16x64x64) on the UCF101 dataset (Soomro et al., 2012).

