# OpenReview forum: "VDT: General-purpose Video Diffusion Transformers via Mask Modeling"
_ICLR.cc/2024/Conference — ICLR 2024 poster_

### Official Review · Reviewer_mnca · 2023-10-27

**Soundness:** 3 good
**Presentation:** 3 good
**Contribution:** 3 good
**Rating:** 6
**Confidence:** 4

**Summary:**

This work introduces a video diffusion model based on the Transformer architecture, a departure from the commonly used U-Net structure in previous Video Diffusion Models (VDM). The utilization of the Transformer model for this task is a significant contribution, especially given its novelty in the field. Additionally, the incorporation of a mask-based modeling mechanism extends the applicability of the model to a wider range of tasks.

**Strengths:**

- Pioneering exploration of the Transformer architecture for video diffusion models, marking a valuable starting point.
- Validation of the effectiveness of the proposed architecture and methods across multiple task datasets.
- Achieving state-of-the-art results on several mainstream benchmarks.

**Weaknesses:**

- While the authors conducted validations on various tasks, the ablation experiments in Table 2 are insufficient. Verification in unconditional video generation tasks would enhance the persuasiveness of the study.
- The method is limited to lower resolutions, diminishing its practical applicability.
- I believe it would be valuable to compare training and inference times. Understanding the computational costs associated with both training and real-time usage is crucial for assessing the feasibility of the approach in real-world scenarios. It would enhance the completeness of the evaluation and provide readers with a comprehensive understanding of the method's efficiency. Therefore, I recommend including a comparison of training and inference times in the manuscript to strengthen the overall analysis.

**Questions:**

- I previously reviewed an earlier version of this manuscript, and there have been notable improvements, particularly in clarifying the significance of the Transformer in the introduction, which I find valuable.
- I am still curious about why the patch-based method is limited to such low resolutions. Even at 256 resolution, the token length is not excessively long, and it could potentially be experimented with. Is this limitation due to computational constraints?
- Additionally, it would be beneficial to compare the proposed method with more recent publications. A comparison with the results summarized in [a] could be enlightening.
- In conclusion, I find the innovation in this paper substantial, and it presents a compelling argument. For now, I am inclined to recommend acceptance.

[a] Xing, Zhen, et al. "A Survey on Video Diffusion Models." arXiv preprint arXiv:2310.10647 (2023).

---

> ### Author Response · Authors · 2023-11-17
> **Response to Reviewer mnca (Part 1)**
>
> Thank you for the positive comments and insightful suggestions. Your insightful questions and valuable suggestions have been immensely helpful in enhancing the paper's quality.
>
> **Q1: While the authors conducted validations on various tasks, the ablation experiments in Table 2 are insufficient. Verification in unconditional video generation tasks would enhance the persuasiveness of the study.**
>
> **A:** Thank you for the suggestion! We have conducted a detailed ablation study on UCF101 using DiT-S. The results show that reducing the Patchsize, increasing the number of Layers, and increasing the Hidden Size all further improve the model's performance. The position of Temporal and Spatial attention and the number of attention heads do not significantly affect the model's results. Further comprehensive analysis indicates that generally, **an increase in GFlops leads to better results, demonstrating the scalability of VDT**. When maintaining the same GFlops, some trade-offs in design are necessary, but overall, the model's performance does not differ significantly. And we have included these experiments in the Appendix A of our paper.
>
> | Patch Size, Iteration=40k| GFlops | UCF101 FVD |
> |------------|------------|------------|
> | 4    |   1.9   | 643.9      |
> | 2    |   7.7   | 554.8      |
> | 1    |   30.9   | **466.2**      |
>
>
> | Head, Iteration=40k| GFlops | UCF101 FVD |
> |------------|------------|------------|
> | 3      |   7.7      | 559.8      |
> | 6      |   7.7      | **554.8**      |
> | 12     |   7.7       | 598.7      |
>
> | Depth, Iteration=40k| GFlops | UCF101 FVD |
> |------------|------------|------------|
> | 6       |   3.9     | 580.7      |
> | 12      |   7.7      | 554.8      |
> | 18      |   11.6     | **500.6**      |
>
>
> | Hidden Size, Iteration=40k| GFlops | UCF101 FVD |
> |------------|------------|------------|
> | 192     |   1.9         | 704.2    |
> | 384     |   7.7         | 554.8      |
> | 768     |   30.9         | **464.2**      |
>
>
> | Architecture, Iteration=40k| GFlops | UCF101 FVD |
> |------------|------------|------------|
> | Spatial First   |   7.7         | 550.2      |
> | Temporal First   |   7.7         | 554.8      |
>
>
> **Q2: The method is limited to lower resolutions, diminishing its practical applicability. I am still curious about why the patch-based method is limited to such low resolutions. Even at 256 resolution, the token length is not excessively long, and it could potentially be experimented with. Is this limitation due to computational constraints?**
>
>
> **A:** Thank you for your question regarding the resolution constraints of our patch-based method. We understand your curiosity, and the limitations are indeed primarily due to data availability and computational resources:
>
> 1. **Data Availability:** The development of high-resolution models (e.g., Align Your Latents) often require access to large and proprietary datasets, which are not publicly available now. Unfortunately, there is a notable scarcity of high-quality, high-resolution video datasets in the public domain. This lack of available data poses a significant challenge in training and testing our method at higher resolutions.
>
> 2. **Computational Constraints:** Addressing the computational aspect, for a video with a resolution of 16x256x256, post VAE encoding, the token length reaches approximately 16,384 (16x32x32). If we were to scale up to a resolution of 16x512x512, the token length would increase to around 65,536 (16x64x64). While our computational resources might be capable of handling this increased scale, it would considerably impact our Batchsize. This factor is crucial for the efficiency and practicality of diffusion models. It's noteworthy that even recent high-resolution works, such as 'Imagen Video' and 'Align Your Latents', still rely heavily on super-resolution (SR) models to manage the computational demands associated with higher resolutions.
>
> These factors combine to create a practical ceiling on the resolution we can effectively work with, given current public data availability and our computational resources. While we acknowledge this limitation, we believe our approach still offers valuable insights and applications within its operational resolution range.

---

> ### Author Response · Authors · 2023-11-17
> **Response to Reviewer mnca (Part 2)**
>
> **Q3: I believe it would be valuable to compare training and inference times. Understanding the computational costs associated with both training and real-time usage is crucial for assessing the feasibility of the approach in real-world scenarios. It would enhance the completeness of the evaluation and provide readers with a comprehensive understanding of the method's efficiency. Therefore, I recommend including a comparison of training and inference times in the manuscript to strengthen the overall analysis.**
>
> **A:** Good suggestion! We have therefore included a comparison of training and inference times (per second), all experiments are conducted on NVIDIA A100 GPU. And we have included these experiments in the Appendix B of our paper.
>
> |   | Resolution | VAE (per sample, bs=1) | Training (per sample, bs=1) | Inference (per sample, t=256, bs=1) |
> |---|------------|------------------------------|-----------------------------|-----------------------------------|
> | VDT-S | 16x64x64      | 0.0042                | 0.022                       | 1.10                                      |
> | VDT-L | 16x64x64      | 0.0042                | 0.051                       | 2.57                              |
> | VDT-S | 16x128x128        | 0.0051                | 0.024                       | 1.21                              |
> | VDT-L | 16x128x128        | 0.0051                | 0.057                       | 6.3                               |
> | VDT-S | 16x256x256        | 0.0058                | 0.026                       | 2.63                              |
> | VDT-L | 16x256x256        | 0.0058                | 0.111                       | 25.31                             |
>
> **Q4: Additionally, it would be beneficial to compare the proposed method with more recent publications. A comparison with the results summarized in [a] could be enlightening.**
>
> **A:** Thank you for the recommendation! we have updated our manuscript to include these comparisons. Should there be any specific recent publications that we may have overlooked or that you believe should be included in our manuscript or any other questions, please feel free to let us know.
>
>
> Thanks again for your time and effort!

---

> ### Author Response · Authors · 2023-11-22
> **Looking forward to your post-rebuttal feedback**
>
> Dear Reviewer mnca,
>
> Thanks again for your constructive suggestions, which have helped us improve the quality and clarity of the paper!
>
> In our previous response, we have carefully studied your comments and made detailed responses. As the discussion period will end soon in 1 days, we are happy to provide any additional clarifications that you may need. Please do not hesitate to contact us if there are other clarifications we can offer. We appreciate your suggestions.
>
> Thanks for your time and efforts!
>
> Warm regards, \
> Authors

---

> > ### Comment · Reviewer_mnca · 2023-11-23
> >
> > Thanks for your reply, most of my concerns solved, I would like to keep the positive score.

---

> ### Author Response · Authors · 2023-11-23
>
> Dear Reviewer mnca,
>
> Thanks again for spending a huge amount of time on our paper, which has helped us improve the quality and clarity of the paper! We are glad to see that our response has addressed most of your concerns.
>
> Thanks for your time and efforts again!
>
> Best, \
> Authors

---

### Official Review · Reviewer_cvvZ · 2023-10-29

**Soundness:** 3 good
**Presentation:** 3 good
**Contribution:** 2 fair
**Rating:** 6
**Confidence:** 4

**Summary:**

This paper introduces Video Diffusion Transformer (VDT), a video generation model with pure transformer architecture.
The core idea is to replace the unet structure of the existing video generation model with a pure transformer structure.
The authors claimed that it's the first successful model in transformer-based video diffusion.
They propose a unified spatial-temporal mask modeling mechanism for VDT, enabling it to unify a diverse array of general-purpose tasks.
They show the effectiveness of VDT on several video-related tasks.

**Strengths:**

1. Compared with the current U-net-based network with demonic attention structures, VDT is a pure transformer network. If it is indeed the first Transformer-only video diffusion network, it would be a good baseline for this area.
2. This paper further proposes the spatiotemporal mask mechanism, which can make VDT adapt to various video tasks, including video generation, video prediction, image-to-video generation, video completion, etc.
3. This paper verifies the effectiveness of VDT on multiple video generation tasks.
4. The code in this article is open source and will help others replicate this work.

**Weaknesses:**

1. The core idea of this paper is simple and reasonable. Replacing the original CNN with transformer is an effective method that has been widely verified in the whole field. In addition, this paper adopts the basic attention mechanism, and the mask strategy proposed is also commonly used in video tasks. So, from this perspective, the contribution is slightly less obvious.
2. I feel that the work in this paper can be a good benchmark in this field. However, this paper only verifies that it can be effective on multiple video tasks. As a good benchmark, this paper should give more detailed experimental analysis of ablation. For example, why does space-time pay attention to time before space? In this paper, the influence of each module on the video generation effect, the comparison experiment of hyperparameter and so on.
3. Tab.5, 142.3 (blacken) is not the best one.

**Questions:**

see weakness

---

> ### Author Response · Authors · 2023-11-17
> **Response to Reviewer cvvZ**
>
> Thank you for the positive comments and insightful suggestions. Your insightful questions and valuable suggestions have been immensely helpful in enhancing the paper's quality.
>
> **Q1: The core idea of this paper is simple and reasonable. Replacing the original CNN with transformer is an effective method that has been widely verified in the whole field. In addition, this paper adopts the basic attention mechanism, and the mask strategy proposed is also commonly used in video tasks. So, from this perspective, the contribution is slightly less obvious.**
>
> **A:** Thank you for your comments on the core idea of our paper. We appreciate your perspective regarding the simplicity and effectiveness of our VDT design.
>
> *  We concur that our Video Diffusion Transformer (VDT) design, while simple, is influenced by established methodologies in the field. Nevertheless, we firmly believe that the essence of innovation often lies in simplicity and effectiveness. Our aim with VDT was not to overcomplicate the design merely for the appearance of novelty, but rather to focus on functional efficiency and adaptability. VDT marks a pioneering effort in applying transformer structures specifically for video diffusion tasks, which, despite its conceptual simplicity, represents a significant and non-trivial leap in this domain.
>
> *  Meanwhile, one of the key strengths of VDT is its ability to unify a broad spectrum of video generation tasks within a single, scalable model, without the need for specialized or task-specific designs. This aspect of VDT, which goes beyond the complexity of its structure, is one import contribution - offering a versatile and robust framework for general purpose video generation tasks.
>
> *  Furthermore, VDT's token-concat approach for conditioning can be seen as  coarse-grained training strategy for autoregressive models. Given the proven success of autoregressive approaches in other domains, VDT's methodology could potentially spearhead significant advancements in large-scale computer vision models.
>
> We agree that the structural design of VDT might not appear novel in isolation. However, the application of this concept in the realm of video diffusion is innovative. The value of our work lies not only in its design but also in its application, scalability, and the potential it offers for video generation using transformer architectures.
>
> **Q2: I feel that the work in this paper can be a good benchmark in this field. However, this paper only verifies that it can be effective on multiple video tasks. As a good benchmark, this paper should give more detailed experimental analysis of ablation. For example, why does space-time pay attention to time before space? In this paper, the influence of each module on the video generation effect, the comparison experiment of hyperparameter and so on.**
>
> **A:** Thank you for the suggestion! We have conducted a detailed ablation study on UCF101 using DiT-S. The results show that reducing the Patchsize, increasing the number of Layers, and increasing the Hidden Size all further improve the model's performance. The position of Temporal and Spatial attention and the number of attention heads do not significantly affect the model's results. Further comprehensive analysis indicates that generally, **an increase in GFlops leads to better results, demonstrating the scalability of VDT**. When maintaining the same GFlops, some trade-offs in design are necessary, but overall, the model's performance does not differ significantly. And we have included these experiments in the Appendix A of our paper.
>
> | Patch Size, Iteration=40k| GFlops | UCF101 FVD |
> |------------|------------|------------|
> | 4    |   1.9   | 643.9      |
> | 2    |   7.7   | 554.8      |
> | 1    |   30.9   | **466.2**      |
>
>
> | Head, Iteration=40k| GFlops | UCF101 FVD |
> |------------|------------|------------|
> | 3      |   7.7      | 559.8      |
> | 6      |   7.7      | **554.8**      |
> | 12     |   7.7       | 598.7      |
>
> | Layer, Iteration=40k| GFlops | UCF101 FVD |
> |------------|------------|------------|
> | 6       |   3.9     | 580.7      |
> | 12      |   7.7      | 554.8      |
> | 18      |   11.6     | **500.6**      |
>
>
> | Hidden Size, Iteration=40k| GFlops | UCF101 FVD |
> |------------|------------|------------|
> | 192     |   1.9         | 704.2    |
> | 384     |   7.7         | 554.8      |
> | 768     |   30.9         | **464.2**      |
>
>
> | Architecture, Iteration=40k| GFlops | UCF101 FVD |
> |------------|------------|------------|
> | Spatial First   |   7.7         | 550.2      |
> | Temporal First   |   7.7         | 554.8      |
>
>
> **Q3: Tab.5, 142.3 (blacken) is not the best one.**
>
> **A:** Thank you for pointing it out! We have revised it.
>
>
> Thanks again for your time and effort! For any other questions, please feel free to let us know during the rebuttal window.

---

> > ### Comment · Reviewer_cvvZ · 2023-11-22
> >
> > Thanks for your detailed response. I would like to give a score that is above borderline considering the contribution.

---

> > > ### Author Response · Authors · 2023-11-22
> > >
> > > Dear Reviewer cvvZ,
> > >
> > > Thanks again for spending a huge amount of time on our paper, which has helped us improve the quality and clarity of the paper! We are glad to see that our response has addressed most of your concerns.
> > >
> > > Thanks for your time and efforts again!
> > >
> > > Best, \
> > > Authors

---

> ### Author Response · Authors · 2023-11-22
> **Looking forward to your post-rebuttal feedback**
>
> Dear Reviewer cvvZ,
>
> Thanks again for your constructive suggestions, which have helped us improve the quality and clarity of the paper!
>
> In our previous response, we have carefully studied your comments and made detailed responses. As the discussion period will end soon in 1 days, we are happy to provide any additional clarifications that you may need. Please do not hesitate to contact us if there are other clarifications we can offer. We appreciate your suggestions.
>
> Thanks for your time and efforts!
>
> Warm regards, \
> Authors

---

### Official Review · Reviewer_HPRy · 2023-11-01

**Soundness:** 3 good
**Presentation:** 2 fair
**Contribution:** 3 good
**Rating:** 6
**Confidence:** 4

**Summary:**

The authors introduce a new method called VDT for generating videos. In this approach, transformer architecture replaces U-net as the backbone for the diffusion model. The method employs a unified spatial-temporal mask for diverse tasks and yields state-of-the-art results. The authors also provide an explanation of the mechanism behind VDT for better understanding.

**Strengths:**

(1) The method is pretty novel and compelling.
(2) Different tasks are designed to clarify the superiorities.
(3) The analysis of the training strategy is technically sound.

**Weaknesses:**

(1) According to Table 4, the FVD values of the diffusion model pre-trained with U-Net are significantly better than VDT. However, the authors mention GPU resource limitations as a potential issue. I suggest that the authors compare the results of relevant tasks to provide further clarification.
(2) It is unclear why MCVD-concat performs better than VDT in FVD.
(3) Due to the resolution, it is difficult to distinguish the differences between Videofusion and VDT in the TaiChi-HD section.

**Questions:**

--The presentation was found to be easy to follow by the reviewer. However, there were some typos in the paper that caused misunderstandings. For example, in one instance, the paper mentioned that "the input can either be pure noise latent features or a concatenation of conditional and noise latent features." The word "conditional" should have been replaced with "conditions". Also, the paper stated that "the results of convergence speed and sample quality are presented in Figure 3 and Table 2, respectively." However, the actual result was shown in Figure 7.

--Although the current quantitative analysis indicates that VDT performs better, the experimental results may not support this fact. Can you provide further analysis and results with improved quality?

---

> ### Author Response · Authors · 2023-11-17
> **Response to Reviewer HPRy (Part 1)**
>
> Thank you for the positive comments and insightful suggestions. Your insightful questions and valuable suggestions have been immensely helpful in enhancing the paper's quality.
>
> **Q1: According to Table 4, the FVD values of the diffusion model pre-trained with U-Net are significantly better than VDT. However, the authors mention GPU resource limitations as a potential issue. I suggest that the authors compare the results of relevant tasks to provide further clarification.**
>
>
> **A:** Thank you for your positive comments and for suggesting a more detailed comparison regarding the FVD values of our diffusion model in relation to U-Net pre-trained models.
>
> We fully recognize the value of such a comparison in providing a clearer understanding of our model's performance. However, as you rightly pointed out, we face significant limitations due to computational resource constraints. This restricts our ability to perform direct comparisons with highly resource-intensive models like Make-A-Video, which benefits from pre-training on extensive datasets (for instance, 2.3 billion image-text pairs and 20 million image-text pairs).
>
> To circumvent this limitation and still offer a meaningful comparison, we have expanded the analysis in Figure 4 (also presented below) to include additional video diffusion models. This broader comparison set includes models such as Make-A-Video, PyoCo, VideoGen, and VDM, which employ similar structures. Our findings indicate that our VDT model competes favorably with non-pretrained structures like VDM and PyoCo, achieving FVD scores of 225.7 compared to their scores of 295.0 and 310.0, respectively.
>
>
>
> | Diff. based on U-Net with Pre: |            |       |
> |--------------------------------|------------|-------|
> | VideoGen                       | 16×256×256 | 345.0 |
> | Make-A-Video                   | 16×256×256 | 81.3  |
> | **Diff. based on U-Net:**          |            |       |
> | PYoCo                          | 16×64×64   | 310.0 |
> | VDM                            | 16×64×64   | 295.0 |
> | **Diff. based on Transformer:**    |            |       |
> | VDT                            | 16×64×64   | 225.7 |
>
> This comparative analysis underscores the efficiency and potential of the VDT network structure, even in the absence of extensive pre-training.
>
> **Q2: It is unclear why MCVD-concat performs better than VDT in FVD**
>
> **A:** Thank you for your excellent question regarding the performance comparison between MCVD-concat and VDT, specifically in the context of the FVD metric.
> * On the Cityscaps dataset, we observed that while VDT outperforms MCVD in terms of SSIM, it shows a slightly weaker performance in FVD compared to MCVD. The FVD metric primarily assesses how closely the sample space of generated videos resembles the original video space. MCVD, which is specifically tailored for video prediction, creates shorter video clips (3-5 frames) based on previous 2 conditional frames and then concatenates them. This approach allows MCVD to maintain the features of the conditional frame effectively in the spatial domain, thereby giving it an edge in the FVD metric. On the other hand, our VDT model generates the entire video sequence in one go, based on the conditional frame, which slightly impacts its performance in FVD. However, it's important to note that VDT still achieves results very close to MCVD in FVD.
>
> * The SSIM metric, contrasting with FVD, measures the similarity between predicted and target frames for each video, necessitating higher accuracy in video prediction. As MCVD generates only a few frames at a time, it can suffer from issues like color jitter and speed discrepancies across the entire video sequence. Our VDT model, which models the entire prediction process holistically, manages to maintain overall video coherence and accuracy. Consequently, VDT significantly outperforms MCVD in the SSIM metric, demonstrating its strength in ensuring consistency and precision in video prediction.
>
> We hope this explanation clarifies the distinct advantages and limitations of both MCVD-concat and VDT in different evaluation metrics and their respective implications for video prediction performance.

---

> ### Author Response · Authors · 2023-11-17
> **Response to Reviewer HPRy (Part 2)**
>
> **Q3: Due to the resolution, it is difficult to distinguish the differences between Videofusion and VDT in the TaiChi-HD section. Although the current quantitative analysis indicates that VDT performs better, the experimental results may not support this fact. Can you provide further analysis and results with improved quality?**
>
> **A:** Sorry for the challenge in differentiating between Videofusion and VDT in the TaiChi-HD section due to the resolution issue.
>
> To address this, we have uploaded high-resolution comparison images in our appendix **(Figure 8)**, which showcases the differences more clearyly. Both VideoFusion and our VDT can generally produce videos. However, VideoFusion's generated results often exhibit less coherent movements (e.g., shaky hands, making it difficult to discern the shape of the hands), while VDT demonstrates more fluid and dynamic movement. Also, in terms of video details, VideoFusion generates clothing folds that are less natural, whereas the folds in VDT adjust more naturally with the movement of the characters. These demonstrate VDT's superior performance in maintaining details and dynamic motion.
>
>
>
> **Q4: There were some typos in the paper that caused misunderstandings.**
>
> **A:** Thank you for pointing them out! We have revised the paper accordingly.
>
> Thanks again for your time and effort! For any other questions, please feel free to let us know during the rebuttal window.

---

> ### Author Response · Authors · 2023-11-22
> **Looking forward to your post-rebuttal feedback**
>
> Dear Reviewer HPRy,
>
> Thanks again for your constructive suggestions, which have helped us improve the quality and clarity of the paper!
>
> In our previous response, we have carefully studied your comments and made detailed responses. As the discussion period will end soon in 1 days, we are happy to provide any additional clarifications that you may need. Please do not hesitate to contact us if there are other clarifications we can offer. We appreciate your suggestions.
>
> Thanks for your time and efforts!
>
> Warm regards, \
> Authors

---

> ### Author Response · Authors · 2023-11-23
>
> Dear reviewer, thanks again for your detailed review.
>
> Since we are nearing the end of the discussion period today (Nov 22, 2023), we wanted to politely reach out to see if our response has addressed your questions satisfactorily (and we would greatly appreciate it if you could revisit your scores accordingly). We are also happy to answer any other/follow up questions before the deadline.
>
> Thanks for your time and efforts!
>
> Best, \
> Authors

---

### Author Response · Authors · 2023-11-17
**General Response**

We sincerely appreciate all reviewers’ time and efforts in reviewing our paper. We are glad to find that reviewers generally recognized our contributions:

* **Model.** VDT is indeed the first Transformer-only video diffusion network, it would be a good baseline for this area. [cvvZ, mnca] The method is pretty novel and compelling [HPRy]. The proposed spatiotemporal mask mechanism, which can make VDT adapt to various video tasks, including video generation, video prediction, image-to-video generation, video completion, etc [cvvZ, mnca].
* **Experiment.** Validation of the effectiveness of the proposed architecture and methods across multiple task datasets [mnca] with multiple video generation tasks [cvvZ]. Achieving state-of-the-art results on several mainstream benchmarks [mnca].
* **Writing.** The presentation was found to be easy to follow by the reviewer [HPRy].


And we also thank all reviewers for their insightful and constructive suggestions, which help a lot in further improving our paper. In addition to the pointwise responses below, we summarize supporting experiments added in the rebuttal according to reviewers’ suggestions.


**New Experiments.**

* Adding more qualitative comparison results on TaiChi-HD [HPRy].
* Detailed ablation study on patch size, attention head, layer, hidden size and arthicture [cvvZ, mnca].
* Training/Inference time of our VDT [mnca].

We thank all reviewers for their insightful and constructive suggestions, which help a lot in further improving our paper. We hope that our pointwise responses below could clarify all reviewers’ confusion and alleviate all of their concerns. We thank all reviewers’ time again.

Best, \
Authors

---

### Public Comment · ~Jaehoon_Yoo1 · 2024-04-05
**Why TATS is considered as GAN method?**

As the review period has concluded, I understand if these changes cannot be incorporated into the paper at this stage. However, I would greatly appreciate it if the authors could address these points or consider them for future revisions.

In Section 2 and Table 4 of the paper, the authors referred to TATS (Ge et al., 2022) and Taming Transformers (Esser et al., 2021) as GAN-based methods. However, based on my understanding, they should be considered as autoregressive transformers similar to imageGPT (Chen et al., 2020). I suggest the authors clarify this distinction and accurately categorize these methods.

Additionally, it seems that the paper may have missed some related works on video generative models. Specifically, I believe MMVID (Han et al., 2022) and MAGViT (Yu et al., 2023) might be discussed in the paper. Both MMVID and MAGViT focus on multi-task learning in videos, including generative tasks, and are based on bidirectional transformers (e.g. MaskGiT (Chang et al., 2022)). Integrating these works into the Related Work section would provide a more comprehensive overview of the field.

Although the review period has ended, I would be grateful if the authors could address these concerns on OpenReview or consider them for future revisions to enhance the quality and completeness of the paper.

- Long Video Generation with Time-Agnostic VQGAN and Time-Sensitive Transformer (Ge et al., 2022)
- Taming Transformers for High-Resolution Image Synthesis (Esser et al., 2021)
- Generative Pretraining from Pixels (Chen et al., 2020)
- Video Synthesis via Multimodal Conditioning (Han et al., 2022)
- MAGVIT: Masked Generative Video Transformer (Yu et al., 2023)
- MaskGIT: Masked Generative Image Transformer (Chang et al., 2022)

---

### Meta-Review · Area_Chair_8wNy · 2023-12-05

**Metareview:**

The paper introducing Video Diffusion Transformer (VDT) has been consistently rated as 6 by all three reviewers, placing it marginally above the acceptance threshold. This rating reflects a balance between the innovative approach of using transformers in video diffusion models and certain limitations in the depth of experimental analysis and presentation.

**Justification For Why Not Higher Score:**

**Presentation and Clarity Issues:**

Reviewer mnca notes presentation issues, such as typos and unclear expressions, which hinder the paper's comprehensibility.

**Experimental and Contribution Limitations:**

Reviewer HPRy and cvvZ both point out the lack of detailed experimental analysis, particularly in the form of ablation studies and comparisons with recent works. This lack of depth in exploring the model's capabilities and limitations impacts the perceived contribution of the paper.

**Performance Concerns:**

 Reviewer mnca observes that in some instances, VDT does not outperform existing models, raising questions about its consistent superiority.

**Methodological Limitations:**

Reviewer HPRy highlights the limitation of VDT to lower resolutions and the absence of critical comparisons in terms of training and inference times, which are important for practical applicability.

**Justification For Why Not Lower Score:**

**Innovation and Novelty:**

All reviewers acknowledge the novelty of employing a transformer architecture in video diffusion models, which marks a significant shift from traditional U-Net structures.

**Versatility and Applicability:**

The paper demonstrates the adaptability of VDT to various video tasks, highlighting its utility across different applications.

**Open Source Contribution:**

Reviewer  cvvZ commends the availability of the code, which promotes transparency and facilitates further research in the field.

**Reviewer Confidence:**

All reviewers express a confidence level of 4, indicating a thorough and balanced evaluation of the paper, despite acknowledging potential gaps in understanding or context.

---

### Decision · Program_Chairs · 2024-01-16

Accept (poster)